# New taxonomic insights for Brazilian *Syrbatus* Reitter (Coleoptera: Staphylinidae: Pselaphinae), including three new species and their mitochondrial genomes

Angélico Asenjo[1,2], Marcely Valois[1], Robson de Almeida Zampaulo[3], Michele Molina[1], Renato R.M. Oliveira[1], Guilherme Oliveira[1] and Santelmo Vasconcelos[1]

[1] Instituto Tecnológico Vale, Belém, Pará, Brazil
[2] Department of Entomology, Natural History Museum, Universidad Nacional Mayor de San Marcos, Lima, Peru
[3] Speleology Office, Vale, Nova Lima, Minas Gerais, Brazil

## ABSTRACT

Here we present a taxonomic treatment for the Brazilian species of *Syrbatus* (*Reitter, 1882*), including the description of three new species (*Syrbatus moustache* Asenjo & Valois **sp. nov.**, *Syrbatus obsidian* Asenjo & Valois **sp. nov.** and *Syrbatus superciliata* Asenjo & Valois **sp. nov.**) from the Quadrilátero Ferrífero (Minas Gerais, Brazil). In addition, we designated lectotypes for the Brazilian species of species-group 2, *Syrbatus centralis* (*Raffray, 1898*), *Syrbatus hetschkoi* (*Reitter, 1888*), *Syrbatus hiatusus* (*Reitter, 1888*), *Syrbatus transversalis* (*Raffray, 1898*), and *Syrbatus trinodulus* (*Schaufuss, 1887*), besides recognizing the holotype for *Syrbatus brevispinus* (*Reitter, 1882*), *Syrbatus bubalus* (*Raffray, 1898*), and *Syrbatus grouvellei* (*Raffray, 1898*). The mitochondrial genomes (mitogenomes) of the three new species are presented, for which we present the phylogenetic placement among Staphylinidae with previously published data.

## INTRODUCTION

The species of the genus *Syrbatus* (*Reitter, 1882*) are distributed in Africa (89 spp.) and South America (30 spp.) (*Newton & Chandler, 1989*; *Asenjo et al., 2019*), where, to date, 25 species were recorded for Brazil, three for Paraguay and two for Argentina (*Asenjo et al., 2019*; *Newton, 2022*). In Brazil, the genus is known to occur in the states of Santa Catarina (13 spp.), São Paulo (eight spp.), Bahia (three spp.), Rio de Janeiro (two spp.) and Minas Gerais (one sp.) (*Asenjo et al., 2013*). Originally described as a subgenus of *Batrisus* (*Aubé, 1833*) by *Reitter (1882)*, based on the characters of the pronotum with a lateral longitudinal sulcus on each side and median longitudinal sulcus absent, *Syrbatus* was later raised to the genus level by *Raffray (1897)*, who primarily focused on the characters of head of the male, with some foveae, spines, or tubercles, and pronotum with a lateral longitudinal sulcus on each, among the species of these two taxa and

Corresponding authors
Angélico Asenjo, pukara8@yahoo.com
Santelmo Vasconcelos, santelmo.vasconcelos@itv.org

*Arthmius* (*LeConte, 1849*). Afterwards, *Raffray (1904)* used characters of the antenna and head to divide the South American species of the genus into six species-groups. *Park (1942)* later reorganized these subdivisions, and *Jeannel (1949)* designated *Batrisus clypeatus* (*Reitter, 1882*) as the type species of *Syrbatus*. Currently, the genus *Syrbatus* contains two subgenera, *Syrbatidius* Jeannel, 1952, and *Syrbatus* s. str., but many of its species has not been assigned to any of them.

The biology of the genus is poorly known, with its species, in general, being part of the soil fauna, as *Syrbatus demoniacus* (*Raffray, 1898*) and *Syrbatus leleupi* (*Jeannel, 1950*), for instance, that were originally found in tobacco plantations in Brazil (*Raffray, 1898*) and in the entrance a of cave in the Democratic Republic of the Congo (*Jeannel, 1950*), respectively. Interestingly, over the last three decades, several authors have documented cave specimens of undescribed species of *Syrbatus* from Brazil (*Gnaspini & Trajano, 1994*; *Pinto da Rocha, 1995*; *Trajano & Bichuette, 2010*; *Gallão & Bichuette, 2018*).

As taxonomic attributions of small invertebrates solely based on morphology may be challenging, especially for females or immature specimens, associating genetic data with the description of new taxa potentially facilitates the identification of subsequently collected samples, besides enabling the application of objective species delimitation approaches (*Thormann et al., 2016*; *Pentinsaari et al., 2019*; *Sire et al., 2019*). Since the formalization of the DNA barcoding approach by *Hebert et al. (2003)*, mitogenome-based markers have been widely employed to characterize and identify animal species, and the continuous advance in sequencing technologies ever since enabled a more cost-effective application of a wider variety of genes, in addition to the official marker, COX1 (*e.g.*, *Johri et al., 2020*; *Asenjo et al., 2023*; *Wei & Shi, 2023*). In public databases, complete mitogenomes of Coleoptera species are relatively abundant, although several taxa within the order are still underrepresented, as in the case of several Staphylinidae genera without any published sequences, such as *Syrbatus*. Thus, as previously mentioned, it is important to include data of genetic references when describing new taxa, especially those with poorly understood evolutionary relationships.

The aim of this study was to describe three new species of the genus *Syrbatus* from Brazil, *Syrbatus moustache* Asenjo & Valois **sp. nov.**, *Syrbatus obsidian* Asenjo & Valois **sp. nov.** and *Syrbatus superciliata* Asenjo & Valois **sp. nov.**, all of which collected in caves of the Quadrilátero Ferrífero, in the state of Minas Gerais. Additionally, we characterized the mitogenomes of these three species, positioning the newly described taxa in the phylogenetic context of Staphylinidae and Pselaphinae.

## MATERIAL & METHODS

### Field collection

The three species described in this work are associated with siliciclastic caves in the Quadrilátero Ferrífero, one of the most important regions for the conservation of subterranean biodiversity in Minas Gerais. This region comprises a large mosaic of vegetation compositions shaped by the conjunction of topography, lithology, climate, and altitude (*Jacobi & Carmo, 2008*), being inserted in a transition zone between two Brazilian

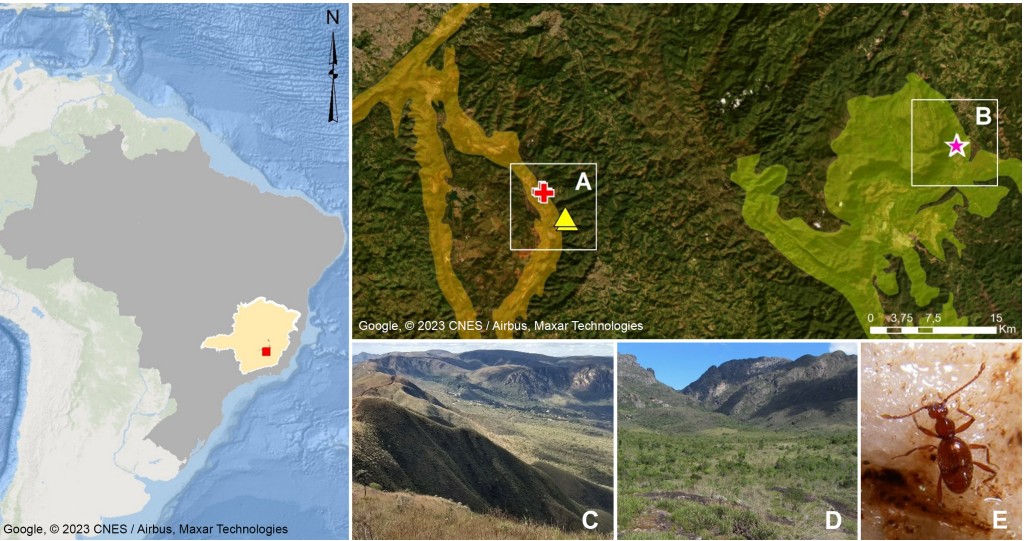

**Figure 1** Geographic distribution of *Syrbatus moustache* Asenjo & Valois sp. nov., *Syrbatus obsidian* Asenjo & Valois sp. nov. and *Syrbatus superciliata* Asenjo & Valois sp. nov. Geographic delimitation of geomorphological units in the Quadrilátero Ferrífero, Minas Gerais, Brazil (Map data: Google, ©2023 CNES/Airbus, Maxar Technologies), showing the occurrence sites of *Syrbatus moustache* Asenjo & Valois **sp. nov.** (yellow triangles) and *Syrbatus superciliata* Asenjo & Valois **sp. nov.** (red crosses) in the mountain range of Serra da Moeda (A and C), and *Syrbatus obsidian* Asenjo & Valois **sp. nov.** (pink stars) in the mountain range of the Escarpa Oriental do Caraça (B and D). Habitus of *Syrbatus superciliata* Asenjo & Valois **sp. nov.** (E).

biodiversity hotspots, the Atlantic Forest and the Cerrado biomes (*Mittermeier et al., 2004*). In general, the climate is characterized as Cwb (subtropical highland climate), with mild and humid summers, and cool and dry winters (*Köppen, 1948*), although it can be strongly influenced by the relief since the average altitude exceeds 1,000 m, with the tallest regions reaching 2× this height. The annual precipitation ranges between 1,250–1,550 mm, with an average annual temperature between 18–19 °C.

The Quadrilátero Ferrífero has an area of approximately 7,200 km² , being considered one of the most important mineral provinces in Brazil, mainly due to its gold and iron deposits. At the same time, the region presents one of the most diverse floras in South America with high rates of endemism (*Giulietti, Pirani & Harley, 1997*), presenting a special biological relevance due to the presence of ferruginous fields and the occurrence of several endemic plant species, besides constituting a unique environment in the country. Being formed by ancient and geologically complex terrains of the Minas Super Group, with varied lithologies (*Alkmim & Marshak, 1998*; *Klein & Ladeira, 2000*), the Quadrilátero Ferrífero encompasses more than two thousand caves currently known in the region (*CECAV, 2021*), and dozens of cave species have been discovered and described in recent years. This large set of caves are distributed in different mountain ranges (geomorphological units), with the species described in this work found in caves in the Serra da Moeda and the Escarpa Oriental do Caraça (Fig. 1). All studied specimens were collected in accordance with the

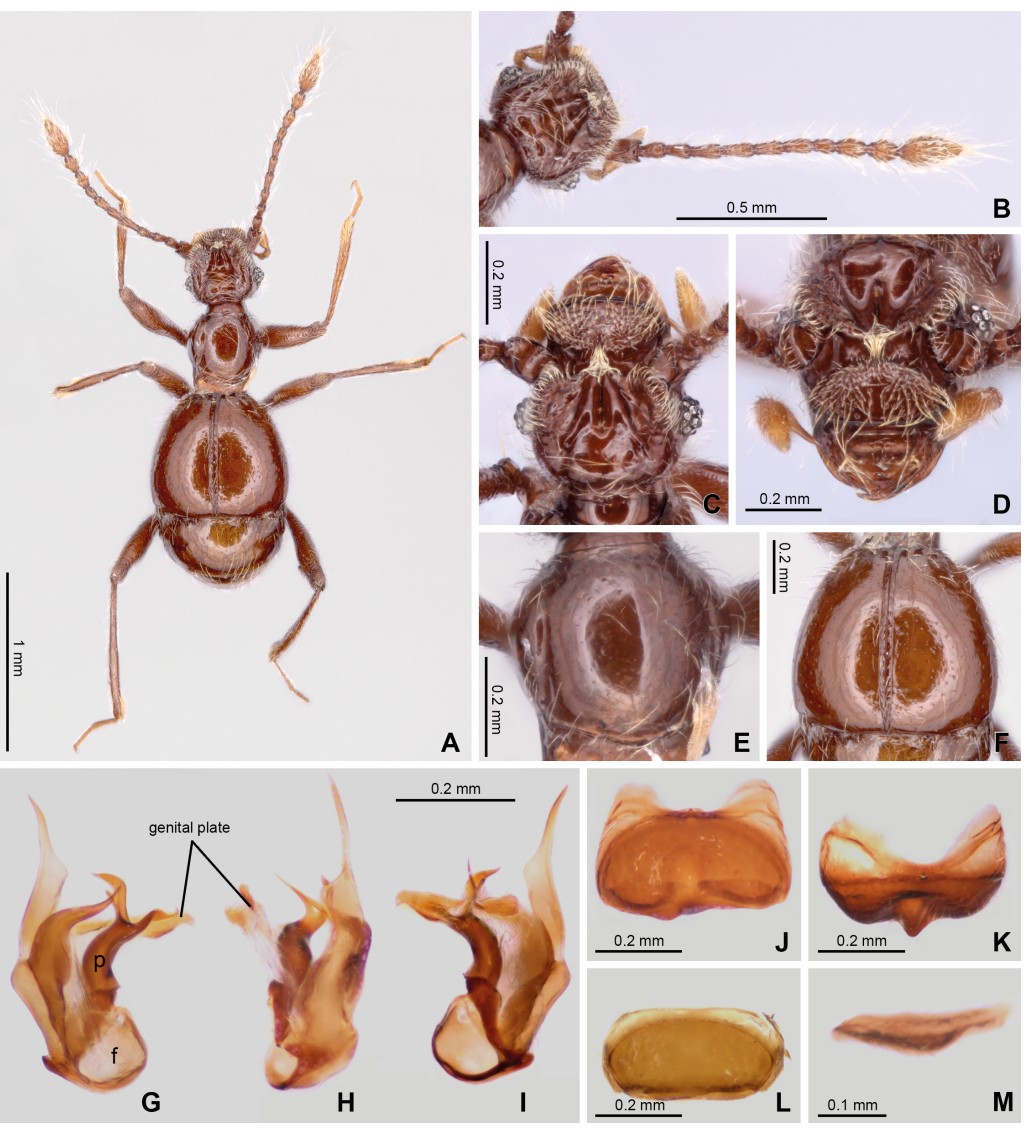

**Figure 2  Habitus and diagnostic characters of holotype male *Syrbatus moustache* Asenjo & Valois sp. nov. (ISLA-110319).** Habitus, dorsal view (A); antenna (B); head, dorsal view (C); head, frontal-dorsal view (D); pronotum (E); elytra (F); aedeagus (G–I); sternum VIII (J–K); tergum VIII (L); sternite IX (M). f, foramen; p, paramere.

sampling permits 065/2013 NUFAS/MG and 424.033/2018, granted by IBAMA/MMA and SEMAD/MG, respectively.

## Morphological analyses

*Specimens.* The apical segments of the abdomen were cleared in a double boiler using 10% KOH for 3 min. Dissections were made under a Zeiss Discovery V12 (4×−125×) stereomicroscope. Pictures were obtained using an AxioCam 506 (ZEISS, Oberkochen, Germany) connected to an Axio Zoom V16 (ZEISS) stereomicroscope, and Photoshop CC 2021 was used for image processing. Morphological character terminology, including

foveation and its abbreviation followed *Chandler (2001)*. All measurements were based using a ZEISS Discovery V12 (4×−125×) stereomicroscope, and the width/length ratios were acquired using the widest and longest parts of the respective structures, being presented in millimeters, based on the holotype. In addition to the data previously described for the holotypes, we provided measurements for the paratypes (Data S1).

Measurements symbols:

BL: body length (from margin of clypeus to posterior margin of tergite VIII)
BW: body width (maximum width of elytra)
EL: elytral length (maximum)
EW: elytral width (maximum)
HL: head length (from anterior margin of clypeus to posterior margin of head disc)
HW: head width (maximum, including eyes)
NW: neck width (minimum)
PL: pronotum length (maximum)
PW: pronotum width (maximum)
In the type label data, quotation marks (" ") separate different labels, and a slash (/) separates different lines within a label. Text within square brackets [ ] is explanatory and is not included on the original labels.

## Depositories

The specimens examined in this revision are deposited in the following collections (curators in parenthesis):

CEMT—Setor de Entomologia da Coleção Zoológica da Universidade Federal de Mato Grosso, Departamento de Biologia e Zoologia, Cuiabá, Mato Grosso, Brazil (Fernando Vaz-de-Mello).

ISLA—Coleção de Invertebrados Subterrâneos de Lavras, Setor de Zoologia, Departamento de Biologia, Universidade Federal de Lavras, Lavras, Minas Gerais State, Brazil (Rodrigo Lopes Ferreira).

ITV—Coleção de DNA do Instituto Tecnológico Vale, Belém, Pará, Brazil (Santelmo Vasconcelos).

MPEG—Museu Paraense Emilio Goeldi, Terra Firme, Belém, Brazil (Orlando Silveira).

MNHN—Muséum National d'Histoire Naturelle, Paris, France (Antoine Mantilleri).

SDEI—Senckenberg Deutschen Entomologischen Institut, Eberswalder, Germany (Mariana Simões).

## Nomenclatural acts

The electronic version of this article in Portable Document Format (PDF) will represent a published work according to the International Commission on Zoological Nomenclature (ICZN), and hence the new names contained in the electronic version are effectively published under that Code from the electronic edition alone. This published work and the nomenclatural acts it contains have been registered in ZooBank, the online registration system for the ICZN. The ZooBank LSIDs (Life Science Identifiers) can be resolved, and

the associated information viewed through any standard web browser. The LSID for this publication is: urn:lsid:zoobank.org:pub:2084B581-B904-486A-B237-0A9D9C839434.

The online version of this work is archived and available from the following digital repositories: PeerJ, PubMed Central and CLOCKSS.

## DNA sequencing, mitogenome assemblies and phylogenetic analysis

A total of seven specimens of the three new species described here were processed to obtain total genomic DNA, being one individual of *Syrbatus moustache* Asenjo & Valois **sp. nov.** [MPEG-01052552/ITV22155 (female)], three of *Syrbatus obsidian* Asenjo & Valois **sp. nov.** [ISLA-110327/ITV22166 (female), ISLA-110326/ITV22168 (male), and MPEG-01052558/ITV22169 (female)], and three of *Syrbatus superciliata* Asenjo & Valois **sp. nov.** [ITV10831 (female), ITV10835 (female), and MPEG-01052550/ITV39056 (female)]. The DNA samples were obtained using the DNeasy Blood & Tissue kit (Qiagen, Valencia, CA, USA), following the manufacturer's protocol for insect samples, subsequently being deposited at the DNA bank of the Instituto Tecnológico Vale (ITV).

Paired-end libraries were constructed from ∼5 ng of the obtained genomic DNA using the Illumina DNA Prep kit (Illumina, Inc., San Diego, CA, USA), following the manufacturer's instructions for low-input samples. After the final purification step, the resultant libraries were quantified with a Qubit 3.0 (Invitrogen, Waltham, MA, USA) fluorimeter, using the Qubit dsDNA High Sensitivity kit (Invitrogen), and analyzed for fragment sizes in a 4200 TapeStation (Agilent Technologies, Santa Clara, CA, USA). Then, the libraries were sequenced in an Illumina NextSeq 2000 platform, with a P1 kit (240 cycles, $2 \times 120$ bp). Resulting raw sequencing reads with base quality < Phred 20 and length <50 bp were trimmed with AdapterRemoval v.2 (*Schubert, Lindgreen & Orlando, 2016*), and the resulting high-quality reads were used to assemble the mitochondrial genomes using NovoPlasty v.4.2 (*Dierckxsens, Mardulyn & Smits, 2017*). Finally, the mitogenomes were annotated with MITOS2 (*Bernt et al., 2013*), with subsequent minor manual corrections using Geneious Prime v.2023 (Biomatters), by comparing the annotated mitogenomes with previously available data for Pselaphinae species.

To position the three new described species in the context of the phylogenetic tree of the family, we obtained all previously published mitogenomes of Staphylinidae species available in the GenBank database (https://www.ncbi.nlm.nih.gov/genbank/) which presented, at least, the complete sequences of the 13 protein coding genes (PCGs), totaling 84 species from 14 subfamilies, plus one species of Hydrophilidae (*Cercyon borealis* Baranowski, 1985) and one of Histeridae (*Euspilotus scissus* (LeConte, 1851)) to be used as outgroups. Amino acid sequences of the 13 PCGs were aligned with MAFFT v7.45 (*Katoh et al., 2002*; Data S2) and maximum likelihood (ML) and Bayesian inference (BI) phylogenetic trees were obtained using RAxML v8 (*Stamatakis, 2014*) and MrBayes v3.2.7 (*Ronquist et al., 2012*), as implemented in the CIPRES portal (http://www.phylo.org/). The ML analysis was performed using the model PROTGAMMA and 1,000 replicates of rapid bootstrapping, and the BI trees were obtained using four simultaneous runs, each with four Markov chains ($T = 0.2$) extended through 20,000,000 generations, sampling every 2,000, and using a burn-in fraction of 25% of the trees. Additionally, to further investigate the differentiation

among the mitogenomes of the three new species, we employed two different lineage delimitation approaches, using the sequences of three specimens of the recently described *Metopiellus crypticus Asenjo et al., 2023*, which was identified as the closest Pselaphinae species among the sampled mitogenomes in the phylogenetic reconstruction. In the first, Assemble Species by Automatic Partitioning (ASAP; *Puillandre, Brouillet & Achaz, 2021*), which uses a genetic distance matrix to identify putative species using a genetic distance matrix, we employed a concatenated matrix of nucleotide sequences of all 13 PCGs as a single locus with the mitogenomes as a single locus, choosing the K80 model ($ts/tv = 2.0$). The second approach, Bayesian Phylogenetics and Phylogeography (BPP; *Yang, 2015*), is based on the multispecies coalescent model under a Bayesian Markov chain Monte Carlo (MCMC) algorithm, for which we provided the nucleotide alignments of the 13 PCGs as independent loci and a guide tree as obtained in the ML phylogenetic reconstruction, using the default parameters.

# RESULTS

## Description

Family Staphylinidae (Latreille, 1802)
Subfamily Pselaphinae (Latreille, 1802)
Tribe Batrisini (*Reitter, 1882*)
Subtribe Batrisina (*Reitter, 1882*)
Genus *Syrbatus* (*Reitter, 1882*)

*Syrbatus moustache*  **Asenjo & Valois, new species**
(Figs. 1, 2 and 3)

urn:lsid:zoobank.org:act:469E3B88-F2C7-4724-9F4C-7A51D1F5FABC

*Type material* (one male, three females). *Holotype:* BRAZIL: male, labeled "BRAZIL: Minas Gerais, /Nova Lima, SM_0046/cave, 29.vii[July]-02.viii[August].2019,/−20.188606, −43.838583 [20°11′18.98″S, 43°50′18.89″W], /Spelayon et al.", "[image data matrix]/Instituto/Tecnológico/Vale/ITV22154", "HOLOTYPE ♂ [red label]/*Syrbatus*/*moustache* sp. nov./Desig. Asenjo et al., 2024" (one male, ISLA-110319). *Paratype:* (three females), labeled: "BRAZIL: Minas Gerais, /Nova Lima, SM_0046/cave, 29.vii[July]-/02.viii[August].2019,/−20.188606, −43.838583 [20°11′18.98″S, 43°50′18.89″W], /Spelayon et al.", "[image data matrix]/Instituto/Tecnológico/Vale/ITV22156" (one female, ISLA-110320). "BRAZIL: Minas Gerais, /Nova Lima, SM_0042/cave, 29.vii[July]-02.viii[August].2019, /−20.192874, −43.836350 [20°11′34.34″S, 43°50′10.86″W], /Spelayon et al.", "[image data matrix]/Instituto/Tecnológico/Vale/ITV22158" (one female, MPEG-01052551). "BRAZIL: Minas Gerais, /Nova Lima, SM_0042/cave, −20.192874, /−43.836350 [20°11′34.34″S, 43°50′10.86″W], 26-31.iii[March].2019, /Spelayon et al.", "[image data matrix]/Instituto/Tecnológico/Vale/ITV22155" (one

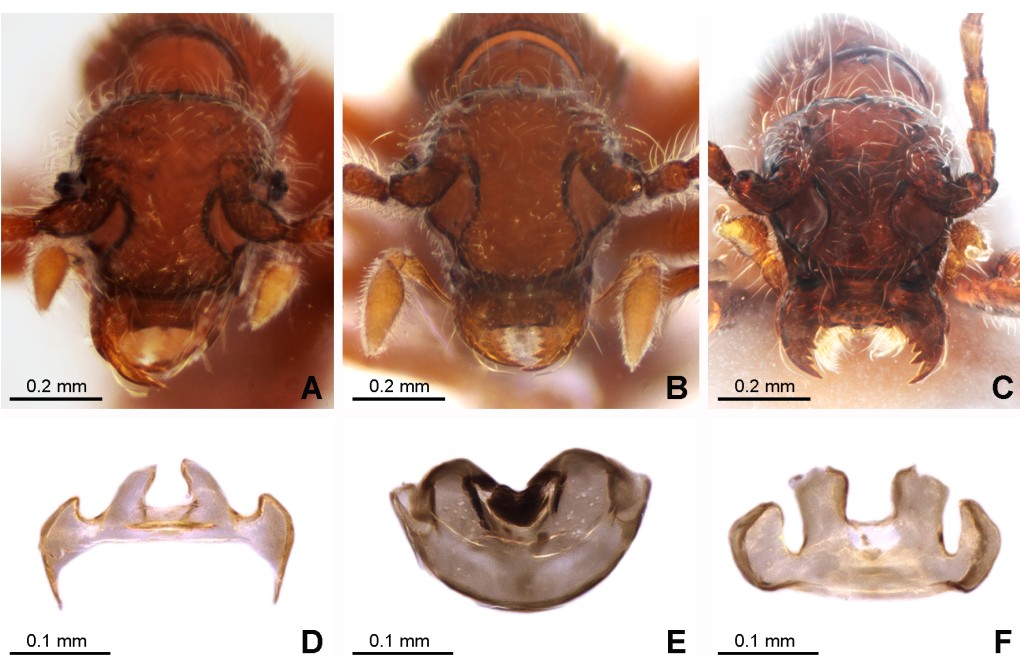

**Figure 3** **Morphology of the female of the three new species of *Syrbatus*.** Frontal-dorsal view of the head (A; ISLA-110320) and genital complex (MPEG-01052551) (D) of *Syrbatus moustache* Asenjo & Valois **sp. nov.**; frontal-dorsal view of the head (B; MPEG-01052554) and genital complex (E; MPEG-01052554) (F) of *Syrbatus obsidian* Asenjo & Valois **sp. nov.**; and frontal-dorsal view of the head (C; MPEG-01052549) and genital complex (F; ISLA-110318) of *Syrbatus superciliata* Asenjo & Valois **sp. nov**.

female, molecular voucher, MPEG-01052552). All paratypes with label "PARATYPE [yellow label]/*Syrbatus*/*moustache* sp. nov./Desig. Asenjo et al., 2024".

*Diagnosis. Syrbatus mustache* Asenjo & Valois **sp. nov.** belongs to species-group 2 (see the Discussion section below), being similar to *Syrbatus superciliata* Asenjo & Valois **sp. nov.**, considering both the habitus and head shape. The main differences between the two are the darker color of the body of *Syrbatus mustache* **sp. nov.**, which is reddish brown (Fig. 2A) (while *Syrbatus superciliata* Asenjo & Valois **sp. nov.** is light brown (Fig. 4A)); head with wide oval transversal (Figs. 2C–2D) area in the anterior region (sling rounded transversal area (Figs. 4C–4D) in *Syrbatus superciliata* Asenjo & Valois **sp. nov.**); posterior margin of sternum VIII with a stout conical projection and its apex rounded (Fig. 2K) in *Syrbatus mustache* **sp. nov.** (thin pointed projection at the middle in with its apex pointed (Fig. 4K) in *Syrbatus superciliata* Asenjo & Valois **sp. nov.**); and aedeagus with thin paramere (Fig. 2G) in *Syrbatus mustache* **sp. nov.** Asenjo & Valois (paramere thick (Fig. 4G) in *Syrbatus superciliata* Asenjo & Valois **sp. nov.**).

Holotype male (Fig. 2A). BL: 2.11. Head, pronotum, elytra and abdomen reddish brown; antennae, mouth parts, tibial apex brown.

Head. Subrectangular (Figs. 2A and 2C–2D), longer (HL: 0.58) than wide (HW: 0.42). Antennal insertions on head not visible in dorsal view. Anterior margin rounded, genal edge rounded and not carinated. Eyes composed of 13 ommatidia and situated at posterior

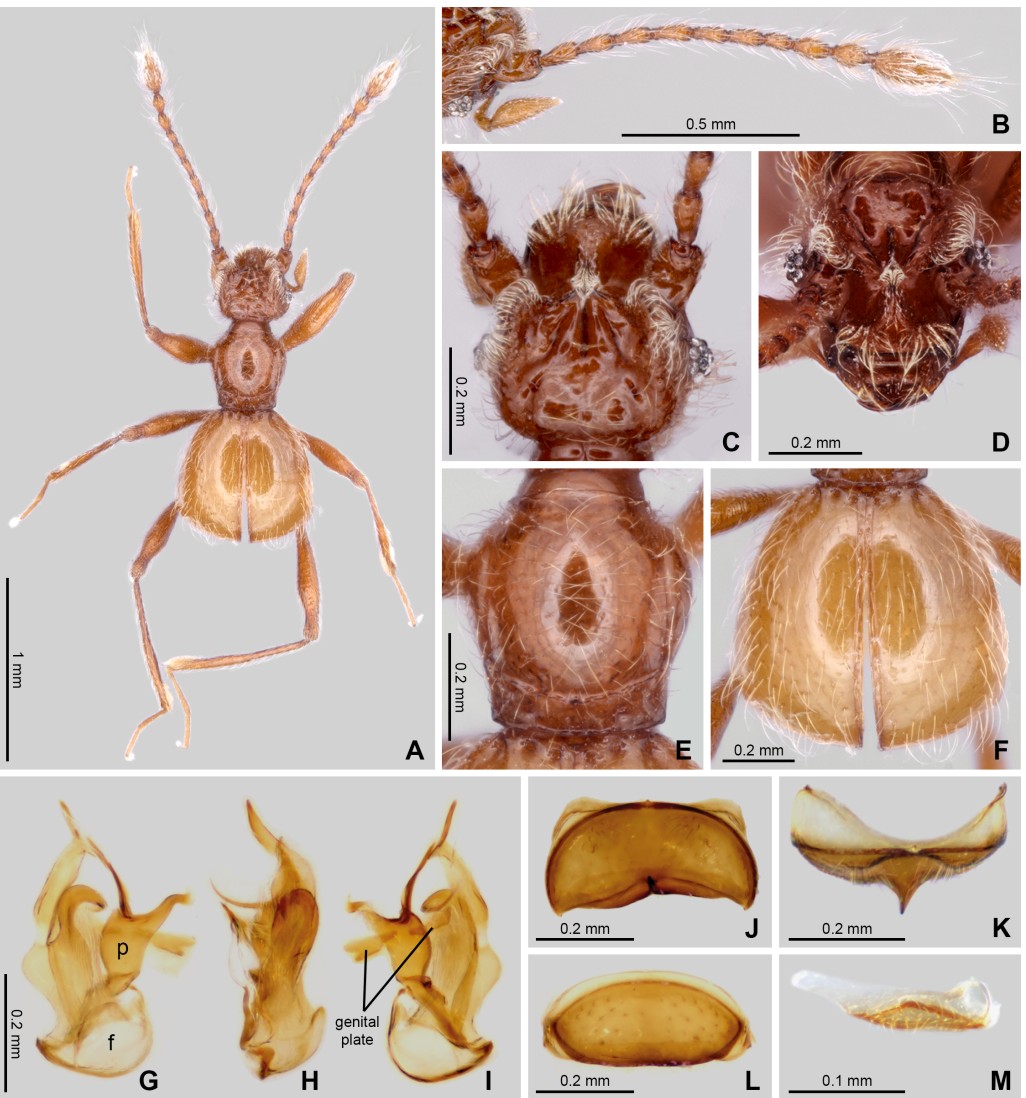

**Figure 4  Habitus and diagnostic characters of holotype male *Syrbatus superciliata* Asenjo & Valois sp. nov. (ISLA-110317).** Habitus, dorsal view (A); antenna (B); head dorsal view (C); head frontal-dorsal view (D); pronotum (E); elytra (F); aedeagus (G–I); sternum VIII (J); sternum VIII (K–L); sternite IX (M). f, foramen; p, paramere.

third of head length. Neck almost one-half width of head (NW: 0.20), with lateral margins rounded. Anterior region of the head with wide oval transversal area, 2× wider than long; and its surface covered with dense setae (Fig. 2C). Frons with long curved transversal slim carina connecting eyes; carina covered with long setae. Both regions cited anteriorly connected by short medial longitudinal keel covered by long setae. The vertex area between eyes with one deep sulcus in format of W-shaped (Fig. 2D) and with vertexal foveae (vf). Ventral surface of head with one gular fovea, without visible gular sulcus. Antennae (Fig. 2A) 2/3 body length. Scape rectangular, deeply impressed medially with margins rounded. Pedicel subclaviform, only slightly shorter than scape. Antennomeres

3–7 cylindrical, gradually broadening to apex, about 2× longer than wide. Antennomere 8 oval, only slightly longer than wide; smaller than the other antennomeres. Antennomeres 9–10 rounded, approximately 2× longer than wide. Antennomere 11 sub-fusiform, 2.5× longer than wide. All antennomeres covered with short and long microsetae.

Thorax: Pronotum hexagonal (Figs. 2A and 2E), slightly longer than wide (PL: 0.45; PW: 0.44) broader at two-third anterior portion and narrower at posterior one-third (Fig. 2F). Surface of pronotum shinning, with long setae direct to center. Pronotal disc distinctly convex in lateral view. Antebasal sulcus (as) present and complete, curved and slender, reaching the posterior ending of the longitudinal sulcus; lateral longitudinal sulcus on each side of pronotum strongly demarcated. Pronotum with anterior margin almost straight and posterior margin convex; with a lateral antebasal fovea (laf), an outer basolateral fovea [oblf], and an inner basolateral fovea (iblf). Posterior angles of pronotum slightly obtuse (Fig. 2E). Prosternum with one lateral procoxal fovea (lpcf). Mesoventrite with one lateral mesosternal foveae (lmsf), one lateral mesocoxal fovea (lmcf), and two median metasternal fovea (mmtf).

Elytra: Trapezoidal (EL: 0.70; EW: 0.80), posterior margin 2× longer than anterior margin (Figs. 2A and 2F). Surface of elytra shining, with long setae sparse and direct posteriorly. Posterior margin curved, with sutural stria (ss) present. Elytron with three basal elytral fovea (bef) at anterior margin. Flight wings extremely reduced (brachypterous species).

Legs: Elongated and slender (Fig. 2A). Femora thickened medially; tibiae slightly widened toward apex, similar in length to femora. Protibiae with microsetae and one apical spur on its apical internal face. Tarsi 3-segmented, with basal tarsomere minute and the two other segments longer. Second segment longer than third. All tarsi with two apical claws; one thicker and other setiform and slender. Pro- and mesocoxae conical, prominent and contiguous. Metacoxae transversely and weakly separated.

Abdomen: With five visible tergites (morphological tergites IV–VIII), posterior margin bordered by fine carina. Tergite IV with a short discal carina (dc) and one basolateral fovea (blf). Sternite IV with two basolateral fovea (blf) and a mediobasal fovea (mbf). Tergite VIII with apex convex. Tergum IX small, subtriangular approximately 8× wider than long (Fig. 2M). Sternum VIII deeply impressed medially, posterior margin sinuous having a conical projection at middle, apex of projection rounded (Figs. 2J–2K). Tergum VIII subrectangular with anterior and posterior margins rounded (Fig. 2L).

Aedeagus: Strongly asymmetric (Figs. 2G–2I), length 0.53 mm; median lobe with large basal capsule and triangular basal foramen; the median lobe strongly bifurcated, with both branches twist. Paramere fused to median lobe to with apex divided into two long curved prolongation. The genital plate divided in two small elongated plates, both attached to aedeagus by a membrane almost transparent.

*Female.* It differs from the male by frons of head with not distinctly expanded over antennal insertions and the absence of excavation, oblique impressions, or cephalic keel (Fig. 3A). The sternum VIII lacking modification such as projections or impressions. Genital complex (Fig. 3D) transversal with lateral corns elongated; anterior border with two central prolongations and one small prolongation on each side.

*Etymology.* The specific epithet name "*moustache*" refers to long setae on the anterior region of front. This is a noun in apposition.

*Distribution.* Known only from Nova Lima, Minas Gerais, Brazil (Fig. 1).

*Syrbatus obsidian*  **Asenjo & Valois, new species**
(Figs. 1, 3 and 5)

urn:lsid:zoobank.org:act:C42D432B-4893-4324-A859-1BD40E47E6D7

*Type material* (eight males, six females). *Holotype:* BRAZIL: male, labeled: "BRAZIL: Minas Gerais, /Catas Altas, FZ_0054/cave, 04.ix[September].2019, /−20.116420, −43.417677 [20°6′59.11″S, 43°25′3.63″W], /Spelayon et al.", "[image data matrix]/Instituto/Tecnológico/Vale/ITV22170", "HOLOTYPE ♂ [red label]/*Syrbatus*/*obsidian* sp. nov./Desig. Asenjo et al., 2024" (MPEG-01052553). *Paratype:* (seven males, six females), the first label as the holotype; "[image data matrix]/Instituto/Tecnológico/Vale/ITV22172" (one male, ISLA-110321); "[image data matrix]/Instituto/Tecnológico/Vale/ITV22171" (one female, MPEG-01052554); "[image data matrix]/Instituto/Tecnológico/Vale/ITV22173" (one female, ISLA-110322). "BRAZIL: Minas Gerais, /Catas Altas, FZ_0050/cave, 04.ix[September].2019, /−20.117065, −43.416092 [20°7′1.43″S, 43°24′57.93″W], /Spelayon et al.", "[image data matrix]/Instituto/Tecnológico/Vale/ITV22161" (one male, CEMT-00138995). "BRAZIL: Minas Gerais, /Catas Altas, FZ_0050/cave, 04.ix[September].2019, /-20.117065, −43.416092 [20°7′1.43″S, 43°24′57.93″W], /Spelayon et al.", "[image data matrix]/Instituto/Tecnológico/Vale/ITV22162" (one male, MPEG-01052555). "BRAZIL: Minas Gerais, /Catas Altas, FZ_0050/cave, 04.ix[September].2019, /−20.117065, −43.416092 [20°7′1.43″S, 43°24′57.93″W], /Spelayon et al.", "[image data matrix]/Instituto/Tecnológico/Vale/ITV22163" (one male, ISLA-110323). "BRAZIL: Minas Gerais, /Catas Altas, FZ_0050/cave, 04.ix[September].2019, /−20.117065, −43.416092 [20°7′1.43″S, 43°24′57.93″W], /Spelayon et al.", "[image data matrix]/Instituto/Tecnológico/Vale/ITV22164" (one male, ISLA-110324). "BRAZIL: Minas Gerais, /Catas Altas, FZ_0050/cave, 04.ix[September].2019, /−20.117065, −43.416092 [20°7′1.43″S, 43°24′57.93″W], /Spelayon et al.", "[image data matrix]/Instituto/Tecnológico/Vale/ITV22165" (one male, ISLA-110325). "BRAZIL: Minas Gerais, /Catas Altas, FZ_0053/cave, 04.ix[September].2019, /−20.116519, −43.417599 [20°6′59.46″S, 43°25′3.35″W], /Spelayon et al.", "[image data matrix]/Instituto/Tecnológico/Vale/ITV22167" (one female, MPEG-01052556). "BRAZIL: Minas Gerais, /Catas Altas, FZ_0049/cave, 04.ix[September].2019, /−20.117099, −43.415814 [20°7′1.55″S, 43°24′56.93″W], /Spelayon et al.", "[image data matrix]/Instituto/Tecnológico/Vale/ITV22159" (one female, CEMT-00138996). "BRAZIL: Minas Gerais, /Catas Altas, FZ_0053/cave, 24.ix[September].2019, /−20.116519, −43.417599 [20°6′59.46″S, 43°25′3.35″W], /Spelayon et al.", "[image data

matrix]/Instituto/Tecnológico/Vale/ITV22166" (one female, molecular voucher, ISLA-110327). "BRAZIL: Minas Gerais, /Catas Altas, FZ_0054/cave, 04.ix[September].2019, /−20.116420, −43.417677 [20°6′59.11″S, 43°25′3.63″W], /Spelayon et al.", "[image data matrix]/Instituto/Tecnológico/Vale/ITV22168" (one male, molecular voucher, ISLA-110326). "BRAZIL: Minas Gerais, /Catas Altas, FZ_0054/cave, 04.ix[September].2019, /−20.116420, −43.417677 [20°6′59.11″S, 43°25′3.63″W], /Spelayon et al.", "[image data matrix]/Instituto/Tecnológico/Vale/ITV22169" (one female, molecular voucher, MPEG-01052558). All paratypes with label "PARATYPE [yellow label]/*Syrbatus*/*obsidian* sp. nov./Desig. Asenjo et al., 2024".

*Diagnosis.* *Syrbatus obsidian* Asenjo & Valois **sp. nov.** belongs to species-group 5 (see the 'Discussion' section). Among the species of species-group 5, *Syrbatus obsidian* Asenjo & Valois **sp. nov.** is similar to *Syrbatus nasutus* (*Reitter, 1888*) in the shape of head and pronotum; *Syrbatus obsidian* Asenjo & Valois **sp. nov.** presents scape moderately dilated (Figs. 5B–5D) (*Syrbatus nasutus* with scape strongly dilated (pl.17, Fig. 10 in *Raffray, 1898*)); anterior margin of head rounded (Fig. 5C) (straight in *Syrbatus nasutus* (pl.17, Fig. 10 in *Raffray, 1898*)); and genal edge rounded (Fig. 5C) (almost straight in *Syrbatus nasutus* (pl.17, Fig. 10 in *Raffray, 1898*)).

Holotype male (Fig. 5A). BL: 1.98 mm. Head, pronotum and abdomen dark brown; antennae, mouth parts legs and elytra reddish brown.

Head: Subtriangular (Figs. 5A and 5C–5D), slightly wider (HW: 0.59) than long (HL: 0.44). Antennal insertions on head not visible in dorsal view. Anterior margin rounded and emarginated at middle, genal edge rounded and carinated. Eyes composed of 19 (right side) and 20 (left side) ommatidia situated at posterior third of head length. Neck almost one-half width of head (NW: 0.23), with lateral margins rounded. Anterior region of head with transversal oval deep excavation, with some dispersed setae (Fig. 5A). The basal margin of excavation with U-shaped carinated and connected to neck by deep longitudinal sulcus. Head with two vertexal foveae (vf). Ventral surface of head with one gular fovea (gf), with indistinct gular sulcus. Antennae (Fig. 5A) 2/3 body length. Scape thick and claviform, apex one-third wider than base, deeply impressed medially and lateral margins strongly margined (Fig. 5B). Pedicel subrectangular, only slightly shorter than scape. Antennomeres 3–7 cylindrical, gradually broadening to apex, about 2× longer than wide. Antennomere 8 oval, only slightly longer than wide; smaller than the other antennomeres. Antennomeres 9–10 rounded, approximately 2× longer than wide. Antennomere 11 sub-fusiform, 2.5× longer than wide. All antennomeres covered by long and short microsetae.

Thorax: Pronotum hexagonal (Figs. 5A and 5E), as long as wide (PL: 0.48; PW: 0.48) broader at two-third anterior portion and narrower at posterior one-third (Fig. 5F). Surface of pronotum shinning, with sparse long setae direct to center. Pronotal disc distinctly convex in lateral view. Antebasal sulcus (as) sinuous and widened medially, reaching the posterior ending of the longitudinal sulcus; and one strong lateral longitudinal sulcus on each side of pronotum. Pronotum with anterior margin convex and posterior margin straight; with a lateral antebasal fovea (laf), an outer basolateral fovea (oblf), and an inner basolateral fovea (iblf). Posterior angles of pronotum straight (Fig. 5E). Prosternum with lateral procoxal

fovea (lpcf). Mesoventrite with lateral mesosternal foveae (lmsf), lateral mesocoxal fovea (lmcf), and two median metasternal fovea (mmtf).

Elytra: (Figs. 5A and 5F). Subrectangular (EL: 0.79; EW: 0.87), apex 2× larger than base. Surface of elytra shinning, with long setae and direct posteriorly; setae sparsely distributed on the elytral disc. Posterior margins curved, with sutural stria (ss) present. Elytron with three basal elytral fovea (bef) at anterior margin. Flight wings extremely reduced (brachypterous species).

Legs: Elongated and slender (Fig. 5A). Femora thickened medially; tibiae slightly widened toward apex, similar in length to femora. Protibiae with microsetae and one apical spur on its internal face. Tarsi 3-segmented, with basal tarsomere minute and the two other segments longer. Second segment longer than third. All tarsi with two apical claws; one thicker and other setiform and slender. Pro- and mesocoxae conical, prominent and contiguous. Metacoxae transversely and weakly separated. Procoxae with small prosternal processed, rounded at apex.

Abdomen: With five visible tergites (morphological tergites IV–VIII) bordered by fine carina. Tergite IV with a short discal carina (dc) and one basolateral fovea (blf). Sternite IV with one basolateral fovea (blf) and a mediobasal fovea (mbf). Tergite VIII with apex convex. Tergum IX triangular approximately 5× wider than long (Fig. 5M). Sternum VIII with a small conical projection at posterior margin and a round impression medially (Figs. 5J–5K). Tergum VIII subrectangular with anterior and posterior margins rounded (Fig. 5L).

Aedeagus (Figs. 5G–5I): strongly asymmetric, length 0.35 mm; median lobe with stout large basoventral projection in the basal capsule and roundly basal foramen; the median lobe strongly bifurcated, in one large and one small branches. Paramere fused to median lobe and broaden to apex. The genital plate large and curved not divided attached to aedeagus by a membrane almost transparent.

*Female.* It differs from the male by frons on head not distinctly expanded over antennal insertions and the absence of excavation, oblique impressions, or cephalic keel (Fig. 3B). The sternum VIII without modifications. Genital complex (Fig. 3E) with semicircular shape; and anterior border with one wide emargination.

Etymology. The specific epithet name "*obsidian*" refers to black light color of body. This is a noun in apposition.

Distribution. Known only from Catas Altas, Minas Gerais, Brazil (Fig. 1).

*Syrbatus superciliata* **Asenjo & Valois, new species**
(Figs. 1, 3 and 4)

urn:lsid:zoobank.org:act:489AB8BC-FF86-4214-AC28-850C99EA37B1

*Type material* (one male, five females). *Holotype:* BRAZIL: male, labeled "BRAZIL: Minas Gerais, /Nova Lima, ABOB_0028/cave, -20.163733, /-43.863682 [20°9′49.44″S,

43°51′49.25″W], 29.iv[April].2019, /R.Zampaulo & M.Simões'', "[image data matrix]/Instituto/Tecnológico/Vale/ITV39053", "HOLOTYPE ♂ [red label]/*Syrbatus*/*superciliata* sp. nov./Desig. Asenjo et al., 2024" (one male, ISLA-110317). *Paratype:* (five females), labeled: "BRAZIL: Minas Gerais, /Nova Lima, ABOB_0028/cave, -20.163733, -43.863682 [20°9′49.44″S, 43°51′49.25″W], /29.iv[April].2019, R.Zampaulo &/M.Simões", "[image data matrix]/Instituto/Tecnológico/Vale/ITV39054" (one female, ISLA-110318). "BRAZIL: Minas Gerais, /Nova Lima, ABOB_0015/cave, 29.iv[April].2019, /-20.165292, -43.861671 [20°9′55.05″S, 43°51′42.01″W], /R.Zampaulo & M.Simões", "[image data matrix]/Instituto/Tecnológico/Vale/ITV39055" (one female, MPEG-01052549). "BRAZIL: Minas Gerais, Nova Lima, ABOB_0015/cave, 01.viii[August].2013, /-20.165292, -43.861671 [20°9′55.05″S, 43°51′42.01″W], /M.Simões", "[image data matrix]/Instituto/Tecnológico/Vale/ITV10831" (one female, molecular voucher, ITV10831). "BRAZIL: Minas Gerais, Nova Lima, ABOB_0015/cave, 01.viii[August].2013, /-20.165292, -43.861671 [20°9′55.05″S, 43°51′42.01″W], /M.Simões, "[image data matrix]/Instituto/Tecnológico/Vale/ITV10835" (one female, molecular voucher, ITV10835). "BRAZIL: Minas Gerais, /Nova Lima, ABOB_0015/cave, 29.iv[April].2019, /-20.165292, -43.861671 [20°9′55.05″S, 43°51′42.01″W], /R.Zampaulo & M.Simões", "[image data matrix]/Instituto/Tecnológico/Vale/ ITV39056" (one female, molecular voucher, ITV39056). All paratypes with label "PARATYPE [yellow label]/*Syrbatus*/*superciliata* sp. nov./Desig. Asenjo et al., 2024".

*Diagnosis.* As mentioned above, *Syrbatus superciliata* Asenjo & Valois **sp. nov.** is, in general, similar to *Syrbatus mustache* Asenjo & Valois **sp. nov.** within the species-group 2. For diagnostical characters, see the diagnosis of *Syrbatus mustache* Asenjo & Valois **sp. nov.** above.

Holotype male (Fig. 4A). BL: 1.52 mm. Head, pronotum and abdomen brown; antennae, mouth parts legs and elytra light brown.

Head: slightly oval (Figs. 4A and 4C–4D), longer (HL: 0.47) than wide (HW: 0.37). Antennal insertions on head not visible in dorsal view. Anterior margin rounded, genal edge rounded and not carinated. Eyes composed of 10 (right side) and 9 (left side) ommatidia situated at posterior third of head length. Neck almost one-half width of head (NW: 0.21), with lateral margins rounded. Anterior region of head with slim rounded transversal area, 3.5× wider than long; surface with very long setae concentrated near margin (Fig. 4C). Eyes connected by long curved transversal slim carina covered of long setae. Both regions cited anteriorly connected by short medial longitudinal keel covered by small setae. The vertex area between eyes with one deep sulcus in format of W-shaped (Fig. 4D) and with vertexal foveae (vf). Ventral surface of head with one gular fovea (gf), without visible gular sulcus. Antennae (Fig. 4B) 2/3 body length. Scape rectangular and slightly stout, deeply impressed medially with margins rounded. Pedicel subclaviform, only slightly shorter than scape. Antennomeres 3–7 cylindrical, gradually broadening to apex, about 2× longer than wide. Antennomere 8 oval, only slightly longer than wide; smaller than the other antennomeres. Antennomeres 9–10 rounded, approximately 2× longer than

wide. Antennomere 11 sub-fusiform, 2.5× longer than wide. All antennomeres covered by long microsetae.

Thorax: Pronotum hexagonal (Figs. 4A and 4E), slightly longer than wide (PL: 0.46; PW: 0.42) broader at medio-anterior portion and narrower at posterior one-third (Fig. 4F). Surface of pronotum shinning, with long setae direct to center. Pronotal disc distinctly convex in lateral view. Antebasal sulcus (as) present and complete, regularly curved and slender, reaching the posterior ending of the longitudinal sulcus on the lateral portions of the pronotal disc. Pronotum with basal and anterior margins weakly convex; with a lateral antebasal fovea (laf), an outer basolateral fovea (oblf), and an inner basolateral fovea (iblf). Posterior angles of pronotum obtuse (Fig. 4F). Prosternum with lateral procoxal fovea (lpcf). Mesoventrite with one lateral mesosternal foveae (lmsf), one lateral mesocoxal fovea (lmcf), and two median metasternal fovea (mmtf).

Elytra: (Figs. 4A and 4F). Subrectangular (EL: 0.75; EW: 0.77), apex 2× larger than base. Surface of elytra shinning, with long setae directed posteriorly, setae sparsely distributed. Posterior margins curved, sutural stria (ss) present. Elytron with three basal elytral fovea (bef) at anterior margin. Flight wings extremely reduced (brachypterous species).

Legs: Elongated and slender (Fig. 4A). Femora thickened medially; tibiae slightly widened toward apex, similar in length to femora. Protibiae with microsetae and one apical spur on its internal face. Tarsi 3-segmented, with basal tarsomere minute and the other two tarsomeres longer. Second tarsomere longer than third. All tarsi with two apical claws; one thicker and other setiform and slender. Pro- and mesocoxae conical, prominent and contiguous. Metacoxae transversely and weakly separated. Protrochanter with small rounded process with long setae.

Abdomen: With five visible tergites (morphological tergites IV–VIII) bordered by fine carina. Tergite IV with a short discal carina (dc) and one basolateral fovea (blf). Sternite IV with two basolateral fovea (blf) and a mediobasal fovea (mbf). Tergite VIII with apex convex. Tergum IX triangular approximately 5× wider than long (Fig. 4M). Sternum VIII deeply impressed medially, posterior margin sinuous having a pointed projection at middle (Figs. 4J–4K). Tergite VIII subrectangular with anterior and posterior margins rounded (Fig. 4L).

Aedeagus (Figs. 4G–4I): strongly asymmetric, length 0.47 mm; median lobe with large basal capsule and roundly basal foramen; the median lobe strongly bifurcated, with both branches twist. Paramere fused to median lobe with apex divided into two long and slim prolongation curved. The genital plate divided in two small elongated plates, both attached to aedeagus by a membrane almost transparent.

*Female.* It differs from the male by frons not distinctly expanded over antennal insertions and the absence of excavation, oblique impressions, or cephalic keel (Fig. 3C). The sternum VIII lacks modifications such as projections or impressions. Genital complex (Fig. 3F) oblong with anterior border with broad two central prolongations and one broad prolongation on each side.

*Etymology.* The specific epithet name "*superciliata*" refers to similar eyebrow above compound eyes. This is a noun in apposition.

*Distribution.* Known only from Nova Lima, Minas Gerais, Brazil (Fig. 1).

*Syrbatus brevispinus* (**Reitter, 1882**)
(Fig. S1)

*Batrisus* (*Syrbatus*) *brevispinus*: *Reitter, 1882*: 137 (original description, drawings pl. 5, Fig. 8. Type locality: "Brasilia: Sao Paolo, 3000 Fufs über dem Meere"), *Reitter, 1888*: 245 (revision).

*Batrisus brevispinus*: *Schaufuss, 1888*: 11 (distribution).

*Arthmius* (*Syrbatus*) *brevispina*: *Raffray, 1898*: 446, 454 (revision), *Raffray, 1904*: 70 (catalogue, species-group 2, and distribution), *Raffray, 1908*: 150 (checklist, distribution), *Raffray, 1911*: 55 (catalogue, distribution).

*Arthmius brevispina*: *Blackwelder, 1944*: 91 (checklist Latin American species, distribution).

*Syrbatus brevispinus*: *Park, 1942*: 234, 237 (revision, species-group 2, and distribution), *Asenjo et al., 2013*: 14 (checklist Brazilian species, distribution).

*Type material* (Fig. S1). "Brésil/Sao. Paulo", "80 Juli/Brasilien/Sao Paolo/3000 FB/leg. [unreadable]", "*Syrbatus*/*brevispina*/Type Reitter/Bresil", "HOLOTYPE/*Syrbatus*/*brevispina* (*Reitter, 1882*)", "MUSÉUM PARIS/1917/COLL. A. RAFFRAY", "S. Brevispina/A. Raffray det.", "HOLOTYPE", "TYPE", "MNHN, Paris/EC21615/[image data matrix]".

*Comments.* *Reitter (1882)*, in the original description, specified that he studied one incomplete female. We studied the only available specimen based on pictures from the Raffray collection with the abdomen absent, probably female, sent by Antoine Mantilleri (photographed by Maéva Pronesti) from the MNHN Entomological Collection with labeled "Type" and "Holotype", and we recognized it as the Holotype.

*Distribution.* Full record and distribution are described in *Asenjo et al. (2013)*.

*Syrbatus bubalus* (**Raffray, 1898**)
(Fig. S2)

*Arthmius* (*Syrbatus*) *bubalus*: *Raffray, 1898*: 444, 454 (original description, drawings pl. 17, Fig. 1. Type locality: "Brésil: Bahia"), *Raffray, 1904*: 70 (catalogue, species-group 2, and distribution), *Raffray, 1908*: 150 (checklist, distribution), *Raffray, 1911*: 55 (catalogue, distribution).

*Arthmius bubalus*: *Blackwelder, 1944*: 91 (checklist Latin American species, distribution).

*Syrbatus bubalus*: *Park, 1942*: 234, 237 (revision, species-group 2, and distribution), *Asenjo et al., 2013*: 14 (checklist Brazilian species, distribution).

*Type material* (Fig. S2). "Bahia", "S. Bubalus/A. Raffray det.", "MUSÉUM PARIS/1917/COLL. A. RAFFRAY", "*Syrbatus* Reitt./*bubalus* Raffr./male symbol type/Bahia", "HOLOTYPE/*Syrbatus*/*bubalus* (*Raffray, 1898*)", "HOLOTYPE", "TYPE", "MNHN, Paris/EC21624/[image data matrix]".

*Comments.* *Raffray (1898)*, in the original description, specified that he studied one male collected by A. Grouvelle [Antoine Henri Grouvelle] from Brazil (Bahia). We

studied one specimen based on pictures from Raffray collection sent by Antoine Mantilleri (photographed by Maéva Pronesti) from the MNHN Entomological Collection with labeled "Type" and "Holotype", and we recognized it as the Holotype.

*Distribution.* Full record and distribution are described in *Asenjo et al. (2013)*.

*Syrbatus centralis* (**Raffray, 1898**)
(Fig. S3)

*Arthmius* (*Syrbatus*) *centralis*: *Raffray, 1898*: 445, 454 (original description. Type locality: "Brésil: Blumenau"), *Raffray, 1904*: 70 (catalogue, species-group 2, and distribution), *Raffray, 1908*: 150 (checklist, distribution), *Raffray, 1911*: 55 (catalogue, distribution).

*Arthmius centralis*: *Blackwelder, 1944*: 91 (checklist Latin American species, distribution).

*Syrbatus centralis*: *Park, 1942*: 234, 237 (revision, species-group 2, and distribution), *Asenjo et al., 2013*: 14 (checklist Brazilian species, distribution).

*Arthmius centralis* Reitter: *Raffray, 1898*: 454 (*in litteris* name).

*Type material* (Fig. S3). "Brésil/Blumenau", "S. centralis/A. Raffray det.", "B. centralis/m. Blūmenaū", "MUSÉUM PARIS/1917/COLL. A. RAFFRAY", "Syrbatus/centralis ♂/Type Reitter/Brésil", "HOLOTYPE/*Syrbatus*/*centralis* (*Raffray, 1898*)", "HOLOTYPE", "TYPE", "MNHN, Paris/EC21623/[image data matrix]".

*Comments. Raffray (1898)*, in the original description, does not specify the number of specimens that he studied, but he, at least, studied one male specimen collected from Brazil (Blumenau). We studied one specimen based on pictures from the Raffray collection sent by Antoine Mantilleri (photographed by Maéva Pronesti) from the MNHN Entomological Collection with labeled "Type" and "Holotype", and it is here designated as Lectotype.

*Distribution.* Full record and distribution are described in *Asenjo et al. (2013)*.

*Syrbatus grouvellei* (**Raffray, 1898**)
(Fig. S4)

*Arthmius* (*Syrbatus*) *grouvellei*: *Raffray, 1898*: 445, 454 (original description, drawings pl. 17, Fig. 3. Type locality: "Brésil: Bahia"), *Raffray, 1904*: 70 (catalogue, species-group 2, and distribution), *Raffray, 1908*: 150 (checklist, distribution), *Raffray, 1911*: 55 (catalogue, distribution).

*Arthmius grouvellei*: *Blackwelder, 1944*: 91 (checklist Latin American species, distribution).

*Syrbatus grouvellei*: *Park, 1942*: 234, 237 (revision, species-group 2, and distribution), *Asenjo et al., 2013*: 15 (checklist Brazilian species, distribution).

*Type material* (Fig. S4). "Bahia", "S. Grouvellei/A. Raffray det.", "MUSÉUM PARIS/1917/COLL. A. RAFFRAY", "'Syrbatus Reitt./grouvellei/Type. Raffr./Bahia.', HOLOTYPE/*Syrbatus*/*grouvellei* (*Raffray, 1898*)", "TYPE", "HOLOTYPE", "MNHN, Paris/EC21622/[image data matrix]"

*Comments. Raffray (1898)*, in the original description, specified that he studied one male collected by A. Grouvelle [Antoine Henri Grouvelle] from Brazil (Bahia). We studied one specimen based on pictures from Raffray collection sent by Antoine Mantilleri (photographed by Maéva Pronesti) from the MNHN Entomological Collection with labeled "Type" and "Holotype", and we recognized it as the Holotype.

*Distribution.* Full record and distribution are described in *Asenjo et al. (2013)*.

*Syrbatus hetschkoi* (**Reitter, 1888**)
(Figs. S5 and S6)

*Batrisus* (*Syrbatus*) *hetschkoi*: *Reitter, 1888*: 245, 250 (original description. Type locality: "Brésil: Bahia").

*Arthmius* (*Syrbatus*) *hetschkoi*: *Raffray, 1898*: 446, 455 (revision, drawings pl. 17, Fig. 8), *Raffray, 1904*: 70 (catalogue, species-group 2, and distribution), *Raffray, 1908*: 150 (checklist, distribution), *Raffray, 1911*: 55 (catalogue, distribution).

*Arthmius hetschkoi*: *Blackwelder, 1944*: 91 (checklist Latin American species, distribution).

*Syrbatus hetschkoi*: *Park, 1942*: 234, 237 (revision, species-group 2, and distribution), *Asenjo et al., 2013*: 15 (checklist Brazilian species, distribution).

*Type material.* First specimen (Fig. S5): "Brésil/Blumenau", "S. Hetschkoi/A. Raffray det.", "MUSÉUM PARIS/1917/COLL. A. RAFFRAY", "Syrbatus Reitt./Hetschkoi Reitt./ ♂ Type/Brésil", "SYNTYPE/*Syrbatus*/*hetschkoi* (*Reitter, 1888*)", "SYNTYPE", "TYPE", "MNHN, Paris/EC21619/[image data matrix]"; second specimen (Fig. S6): "Brésil/Blumenau", "S. Hetschkoi/A. Raffray det.", "MUSÉUM PARIS/1917/COLL. A. RAFFRAY", "SYNTYPE/*Syrbatus*/*hetschkoi* (*Reitter, 1888*)", "SYNTYPE", "MNHN, Paris/EC21620/[image data matrix]".

*Comments. Reitter (1888)*, in the original description, does not specify the number of specimens that he studied; the specimen(s) was(were) collected by Lothar Hetschko from Brazil (Blumenau). We studied, two specimens based on pictures from Raffray collection sent by Antoine Mantilleri (photographed by Maéva Pronesti) from the MNHN Entomological Collection, labeled as "Type" (first specimen (Fig. S5)) and "syntype" (second specimen (Fig. S6)), and it is here designated as Lectotype and Paralectotype, respectively.

*Distribution.* Full record and distribution are described in *Asenjo et al. (2013)*.

*Syrbatus hiatusus* (**Reitter, 1888**)
(Figs. S7 and S8)

*Batrisus* (*Arthmius*) *hiatusus*: *Reitter, 1888*: 247, 257 (original description. Type locality: "Brésil: Blumenau").

*Arthmius* (*Syrbatus*) *hiatusus*: *Raffray, 1898*: 445, 454 (revision), *Raffray, 1904*: 70 (catalogue, species-group 2, and distribution), *Raffray, 1908*: 150 (checklist, distribution), *Raffray, 1911*: 55 (catalogue, distribution), *Bruch, 1914*: 304 (catalog Argentinean species).

*Arthmius hiatusus*: *Blackwelder, 1944*: 91 (checklist Latin American species, distribution).

*Syrbatus hiatusus*: *Jeannel, 1949*: 128 (drawings Fig. 57), *Park, 1942*: 234, 237 (revision, species-group 2, and distribution), *Asenjo et al., 2013*: 15 (checklist Brazilian species, distribution).

*Type material.* First specimen (Fig. S7): "Brésil/Blumenau", "B. hiatusus/m. Blūmenaū", "[red square label]", "MUSÉUM PARIS/1917/COLL. A. RAFFRAY", "S. Hiatúsus/A. Raffray det.", "Syrbatus Reitt./hiatusus Reitt./ ♂ Type/Brésil", "HOLO-TYEPE/*Syrbatus*/*hiatusus* (*Reitter, 1888*)", "HOLOTYPE", "MNHN, Paris/EC21616/[image data matrix]"; second specimen (Fig. S8): "Brésil/Blumenau", "S. Hiatusus/A. Raffray det.", "MUSÉUM PARIS/1917/COLL. A. RAFFRAY", "MNHN, Paris/EC21617/[image data matrix]".

*Comments. Reitter (1888)*, in the original description, mentioned that he studied some specimens, but without specifying the number, collected by Lothar Hetschko from Brazil (Blumenau). We studied two specimens based on pictures from the Raffray collection sent by Antoine Mantilleri (photographed by Maéva Pronesti) from the MNHN Entomological Collection, the first specimen (Fig. S7) labeled "Holotype" and the second specimen (Fig. S8), and it is here designated as Lectotype and Paralectotype, respectively.

*Distribution.* Full record and distribution are described in *Asenjo et al. (2013)*, and for Argentina (Buenos Aires) in *Bruch (1914)*.

*Syrbatus transversalis* (**Raffray, 1898**)
(Fig. S9)

*Arthmius* (*Syrbatus*) *transversalis*: *Raffray, 1898*: 447, 455 (original description. Type locality: "Brésil: S. Antonio"), *Raffray, 1904*: 70 (catalogue, species-group 2, and distribution), *Raffray, 1908*: 150 (checklist, distribution), *Raffray, 1911*: 55 (catalogue, distribution).

*Arthmius transversalis*: *Blackwelder, 1944*: 92 (checklist Latin American species, distribution).

*Syrbatus transversalis*: *Park, 1942*: 234, 237 (revision, species-group 2, and distribution), *Asenjo et al., 2013*: 17 (checklist Brazilian species, distribution).

*Type material* (Fig. S9). "Brésil/St.Antonio", "S.Transversalis/A. Raffray det.", "MUSÉUM PARIS/1917/COLL. A. RAFFRAY", "Syrbatus Reitt./transversalis/Type. Raffr./Brésil", "HOLOTYEPE/*Syrbatus*/*transversalis* (*Raffray, 1898*)", "TYPE", "HOLO-TYPE", "MNHN, Paris/EC21621/[image data matrix]".

*Comments. Raffray (1898)*, in the original description, does not specify the number of specimens that he studied, but he, at least, studied one male specimen collected by S. Antonio from Brazil. We studied one specimen based on pictures from the Raffray collection sent by Antoine Mantilleri (photographed by Maéva Pronesti) from the MNHN

Entomological Collection with labeled ''Type'' and ''Holotype'', and it is here designated as Lectotype.

*Distribution.* Full record and distribution are described in *Asenjo et al. (2013)*.

*Syrbatus trinodulus* (**Schaufuss, 1887**)
(Fig. S10)

*Batrisus* (*Syrbatus*) *trinodulus*: *Schaufuss, 1887*: 145 (original description. Type locality: ''Minas Geraes, Brasilia'').

*Batrisus trinodulus*: *Gaedike, 1984*: 456 (catalogue of type specimens).

*Arthmius trinodulus*: *Schaufuss, 1888*: 17 (distribution).

*Arthmius* (*Syrbatus*) *trinodulus*: *Raffray, 1904*: 71 (catalogue, distribution and unplaced any group), *Raffray, 1908*: 151 (checklist, distribution), *Raffray, 1911*: 55 (catalogue, distribution), *Park, 1942*: 237 (revision, unplaced, and distribution), *Blackwelder, 1944*: 92 (checklist Latin American species, distribution).

*Syrbatus trinodulus*: *Asenjo et al., 2013*: 17 (checklist Brazilian species, distribution).

*Type material* (Fig. S10). ''min geraes [Minas Gerais state in Brazil]'', ''Minas/geraes'', ''Syntypus'', ''Batrisus/(Syrbatus)/3-nodulus.'', ''SDEI Coleoptera/#302700''.

*Comments. Schaufuss (1887)*, in the original description, does not specify the number or sex of the specimens studied. We studied one specimen (unidentified sex) based on pictures sent by Mandy Schröter from the SDEI Entomological Collection with a ''syntypus'' label, and it is here designated as Lectotype. The species is considered unplaced in any species-group, as proposed by *Raffray (1904)*.

*Distribution.* Full record and distribution are described in *Asenjo et al. (2013)*.

## Mitogenomes and phylogenetic placement

All seven assembled mitochondrial genomes [*Syrbatus moustache* Asenjo & Valois **sp. nov.**: OR625193 (ITV22155); *Syrbatus obsidian* Asenjo & Valois **sp. nov.**: OR625194 (ITV22166), OR625195 (ITV22168) and OR625196 (ITV22169); and *Syrbatus superciliata* Asenjo & Valois **sp. nov.**: OR625197 (ITV10831), OR625198 (ITV10835) and OR625199 (ITV39056)] presented the standard structure and gene content of Metazoans, although the genes were arranged in an unusual order, considering the pattern described for all other Staphylinidae species with available mitogenomes (Fig. 6A; Table 1). Most genes were encoded in the light (L) strand in the mitogenomes of the three new species, except for four PCGs (ND1, ND4, ND4L and ND5), the two rRNA genes (rrnL and rrnS), and eight tRNA genes (trnC, trnF, trnH, trnL1, trnP, trnP, trnQ, trnV and trnY), which were observed in the heavy (H) strand (Table 1). Thus, using the L strand as the standard, we observed the following gene order for the three new species: trnM, COX1, trnK, ND3, trnR, trnC, COX3, ND4, trnT, trnP, ND6, CYTB, trnI, trnQ, ND2, trnW, trnY, trnL2, COX2, trnD, ATP8, ATP6, trnG, trnA, trnN, trnS1, trnE, trnF, ND5, trnH, ND4L, trnS2, ND1, trnL1, rrnL, rrnS and trnV. Moreover, since some of the mitogenome assemblies presented different levels of completeness, we could not recover genes in the extremities for some

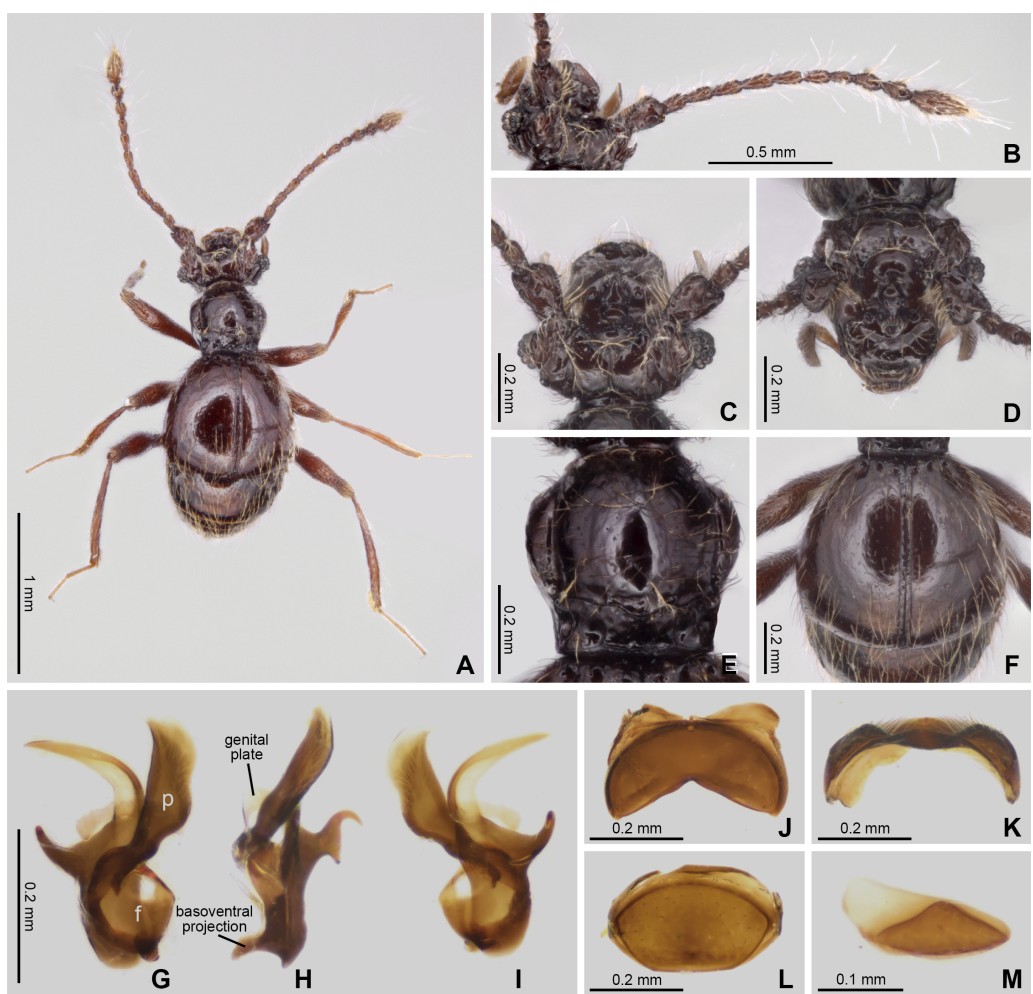

**Figure 5** **Habitus and diagnostic characters of holotype male *Syrbatus obsidian* Asenjo & Valois sp. nov. (MPEG-01052553).** Habitus, dorsal view (A); antenna (B); head, dorsal view (C); head, frontal-dorsal view (D); pronotum (E); elytra (F); aedeagus (G–I); sternum VIII (J–K); tergum VIII (L); sternite IX (M). f, foramen; p, paramere.

specimens, as in the case of all three accessions of *Syrbatus obsidian* Asenjo & Valois **sp. nov.** (OR625194, OR625195 and OR625196), missing part of the rrnL, besides the whole rrnS and trnV genes, and one accession of *Syrbatus superciliata* Asenjo & Valois **sp. nov.** (OR625199), missing part of the rrnL, and the complete sequences of rrnS, trnM and trnV (Table 1).

The mitogenomes of the three new species were largely syntenic, presenting intergenic regions and gene superpositions with similar lengths along the fully aligned extension (from COX1 to mid rrnL, due to the missing parts of some accessions), except for a non-coding intergenic region with 473 bp between trnS2 and NAD1, which was only present in the mitogenomes of *Syrbatus obsidian* Asenjo & Valois **sp. nov.** (Fig. 6B). We also observed similar nucleotide contents between *Syrbatus moustache* Asenjo & Valois **sp. nov.** and *Syrbatus superciliata* Asenjo & Valois **sp. nov.**, with %GC = 17.3 and 17.0, respectively,

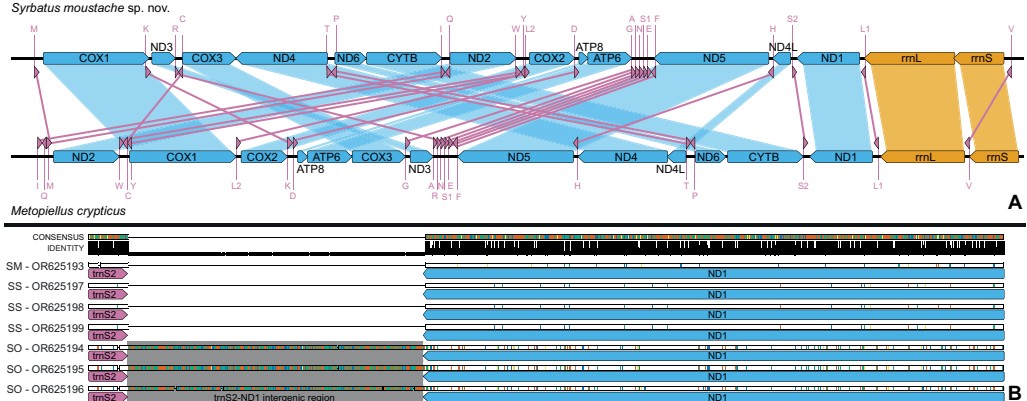

**Figure 6  Mitochondrial genome representation of the three new species of _Syrbatus_, comparing the gene order with _Metopiellus crypticus_.** Comparison of the order of all 37 mitochondrial genes between _Syrbatus moustache_ Asenjo & Valois **sp. nov.** and _Metopiellus crypticus_ (A); colored arrows pointing to the left and right represent the transcription regions of protein coding genes (blue), rRNA genes (orange) and tRNA genes (pink) on the L and H strands, respectively; blue, orange, and pink connections between genes indicate the reorganization of the mitogenome of _Syrbatus moustache_ Asenjo & Valois **sp. nov.** in relation to the gene disposition of _Metopiellus crypticus_, a species presenting the ancestral pattern of insect species. Detail of the region between the genes trnS2 and ND1 in the mitogenomes of _Syrbatus moustache_ Asenjo & Valois **sp. nov.**, SM, _Syrbatus obsidian_ Asenjo & Valois **sp. nov.**, SO, and _Syrbatus superciliata_ Asenjo & Valois **sp. nov.**, SS, evidencing the extended intergenic region in the three accessions of the former; the black bars above the arrows indicate identity levels among mitogenomes (B).

and _Syrbatus obsidian_ Asenjo & Valois **sp. nov.** as the most diverging species, with %GC = 18.9. Overall, the three species shared most of the start codons, which presented the usual ATN configuration, except in the case of COX1 of _Syrbatus obsidian_ Asenjo & Valois **sp. nov.**, which presented TTA, which is an alternative start codon (encoding the amino acid leucine) for invertebrate mitogenomes (Table 1). In addition, only two genes were recorded with diverging stop codons among the three species: COX3, for which _Syrbatus obsidian_ Asenjo & Valois **sp. nov.** presented TAG instead of TAA; and CYTB, with an incomplete stop codon T in _Syrbatus superciliata_ Asenjo & Valois **sp. nov.**, also TAA in the other two species (Table 1).

The phylogenetic reconstruction within Staphylinidae based on ML and BI analyses using a concatenated matrix with amino acid sequences of all 13 PCGs resulted in mostly well-resolved trees, with most of the 14 sampled subfamilies being recovered as well-supported clades (Fig. 7). However, the only two species of Tachiporinae included in the phylogenetic analyses appeared within two distinct lineages, in which _Sepedophilus bipunctatus_ (_Gravenhorst, 1802_) appeared as sister to the clade composed of Steninae and Euaesthetinae species, and _Tachinus subterraneus_ (_Linnaeus, 1758_) was more closely related to Apateticinae and Omaliinae. Also, Paederinae was recovered as paraphyletic, with _Habrocerus capillaricornis_ (_Gravenhorst, 1806_), from Habrocerinae, appearing deeply nested within the former subfamily. On the other hand, while Staphylininae was poorly supported in the ML analysis (BS = 68), the group presented the maximum value of posterior probability in the BI trees (PP = 1).

**Table 1** General features of the mitochondrial genes of *Syrbatus moustache* Asenjo & Valois sp. nov., *Syrbatus obsidian* Asenjo & Valois sp. nov. and *Syrbatus superciliata* Asenjo & Valois sp. nov.

| Gene | Size (bp)[a] (Sm/So/Ss) | Indels (Sm/So/Ss) | Mismatches (Sm/So/Ss) | % Mismatches (Sm/So/Ss) | Coding strand | Start codon | Stop codon |
|---|---|---|---|---|---|---|---|
| ATP6 | 653/638/653 | NA/0/0 | NA/15/1 | NA/2.35/0.15 | L | ATA | TA |
| ATP8 | 138/150/135[b] | NA/0/6 | NA/3/0 | NA/2.00/0.00 | L | ATC | TAA |
| COX1 | 1566/1533/1566 | NA/0/0 | NA/26/0 | NA/1.70/0.00 | L | ATT/TTA/ATT | TAA |
| COX2 | 672/687/672 | NA/0/0 | NA/9/0 | NA/1.31/0.00 | L | ATA | TAA |
| COX3 | 792/783/792 | NA/0/0 | NA/24/3 | NA/3.07/0.38 | L | ATA/ATT/ATA | TAA/TAG/TAA |
| CYTB | 1116/1077/1126 | NA/0/0 | NA/29/1 | NA/2.69/0.09 | L | ATG | TAA/TAA/T |
| ND1 | 927/927/927 | NA/0/NA | NA/22/1 | NA/2.37/0.11 | H | ATT/ATA/ATT | TAA |
| ND2 | 975/978/975 | NA/0/0 | NA/20/0 | NA/2.04/0.00 | L | ATT | TAA |
| ND3 | 351/351/348 | NA/0/0 | NA/4/0 | NA/1.14/0.00 | L | ATT/ATT/ATA | TAA |
| ND4 | 1353/1314/1347 | NA/0/0 | NA/25/0 | NA/1.90/0.00 | H | ATG | TAA |
| ND4L | 256/262/256 | NA/0/0 | NA/4/0 | NA/1.45/0.00 | H | ATA | T |
| ND5 | 1695/1710/1695 | NA/0/0 | NA/36/2 | NA/2.11/0.12 | H | ATT | TAA |
| ND6 | 465/468/465 | NA/0/0 | NA/9/0 | NA/1.92/0.00 | L | ATT | TAA |
| rrnL | 1340/NA[b]/1336 | NA/NA/2 | NA/NA/1 | NA/NA/0.00 | H | NA | NA |
| rrnS | 747/NA[b]/764 | NA/NA/0 | NA/NA/0 | NA/NA/0.00 | H | NA | NA |
| trnA (tgc) | 63/70/63 | NA/0/0 | NA/3/0 | NA/4.29/0.00 | L | NA | NA |
| trnC (gca) | 61/62/61 | NA/0/0 | NA/1/0 | NA/1.61/0.00 | H | NA | NA |
| trnD (gtc) | 64/63/62 | NA/0/0 | NA/0/0 | NA/0.00/0.00 | L | NA | NA |
| trnE (ttc) | 63/63/61 | NA/0/0 | NA/0/0 | NA/0.00/0.00 | L | NA | NA |
| trnF (gaa) | 61/60/61 | NA/0/0 | NA/0/0 | NA/0.00/0.00 | H | NA | NA |
| trnG (tcc) | 60/60/60 | NA/0/0 | NA/0/0 | NA/0.00/0.00 | L | NA | NA |
| trnH (gtg) | 70/63/72 | NA/0/0 | NA/1/0 | NA/1.59/0.00 | H | NA | NA |
| trnI (gat) | 61/62/63 | NA/0/0 | NA/1/0 | NA/1.61/0.00 | L | NA | NA |
| trnK (ctt) | 71/70/71 | NA/0/0 | NA/0/0 | NA/0.00/0.00 | L | NA | NA |
| trnL1 (tag) | 62/67/62 | NA/4/0 | NA/1/0 | NA/1.49/0.00 | H | NA | NA |
| trnL2 (taa) | 64/63/64 | NA/0/0 | NA/0/0 | NA/0.00/0.00 | L | NA | NA |
| trnM (cat) | 68/69/NA | NA/0/NA | NA/1/NA | NA/1.45/NA | L | NA | NA |
| trnN (gtt) | 64/66/63 | NA/0/0 | NA/0/0 | NA/0.00/0.00 | L | NA | NA |
| trnP (tgg) | 70/67/70 | NA/0/0 | NA/3/0 | NA/4.48/0.00 | H | NA | NA |
| trnQ (ttg) | 66/68/66 | NA/0/0 | NA/1/0 | NA/1.47/0.00 | H | NA | NA |
| trnR (tcg) | 53/57/54 | NA/1/0 | NA/2/0 | NA/3.51/0.00 | L | NA | NA |
| trnS1 (tct) | 52/54/52 | NA/0/0 | NA/0/0 | NA/0.00/0.00 | L | NA | NA |
| trnS2 (tga) | 61/61/62 | NA/0/0 | NA/0/0 | NA/0.00/0.00 | L | NA | NA |
| trnT (tgt) | 63/61/63 | NA/0/0 | NA/1/0 | NA/1.64/0.00 | L | NA | NA |

**Table 1** (*continued*)

| Gene | Size (bp) [a] (Sm/So/Ss) | Indels (Sm/So/Ss) | Mismatches (Sm/So/Ss) | % Mismatches (Sm/So/Ss) | Coding strand | Start codon | Stop codon |
|---|---|---|---|---|---|---|---|
| trnV (tac) | 62/NA[b]/63 | NA/NA/NA | NA/NA/NA | NA/NA/NA | H | NA | NA |
| trnW (tca) | 68/67/72 | NA/0/2 | NA/1/0 | NA/1.49/0.00 | L | NA | NA |
| trnY (gta) | 61/64/61 | NA/0/0 | NA/1/0 | NA/1.56/0.00 | H | NA | NA |

**Notes.**

Sequenced mitogenomes of *Syrbatus moustache* Asenjo & Valois sp. nov. (Sm—OR625193), *Syrbatus obsidian* Asenjo & Valois sp. nov. (So—OR625194, OR625195 and OR625196) and *Syrbatus superciliata* Asenjo & Valois sp. nov. (Ss—OR625197, OR625198 and OR625199), indicating the size of the transcription regions, presence of indel events, number of intraspecific mismatches after the alignment of the mitogenomes, coding strand, and sequences of both start and stop codons.

[a]Whenever any of the accessions presented incomplete or missing genes in the mitogenome assembly, we showed the data for those with a complete sequence; also, in the cases with intraspecific variation in the gene sizes, we presented the data of the largest one.

[b]The rrnL, rrnS and trnV genes were either incomplete or missing in the three mitogenomes assembled for *Syrbatus obsidian* Asenjo & Valois sp. nov.

Regarding Pselaphinae, the subfamily was recovered as monophyletic with strong support (BS = 100; PP = 1), forming, with Neophoninae, the sister clade to the remaining Staphylinidae subfamilies (Fig. 7). The three new species of *Syrbatus* formed a fully supported monophyletic group (BS = 100; PP = 1) with well-resolved interspecific relationships, forming, alongside *Metopiellus crypticus*, another clade with maximum statistical support (BS = 100; PP = 1). As also indicated by the morphological similarity, *Syrbatus moustache* Asenjo & Valois **sp. nov.** and *Syrbatus superciliata* Asenjo & Valois **sp. nov.** were more closely related to each other, with a considerably lower interspecific divergence in comparison to the phylogenetic distance of both to *Syrbatus obsidian* Asenjo & Valois **sp. nov.** (Fig. 7).

In addition, we also recovered the three new species as clearly separated clusters in both lineage delimitation approaches. In the ASAP analysis, the best score was obtained for four subsets and a threshold distance of 0.0467 (asap-score = 1.00; $P$-val = 0.0019; $W$ = 0.0387; Table S1), considering the three species of *Syrbatus* and *Metopiellus crypticus*, grouping in the same pattern as observed in the phylogenetic reconstruction. Similarly, the four species were recovered as independent lineages with strong support (PP = 0.9828; Table S1), all of which presenting individual high posterior probabilities in the BPP analysis, being 0.9829 for *Syrbatus moustache* Asenjo & Valois **sp. nov.**, 0.9901 for *Syrbatus obsidian* Asenjo & Valois **sp. nov.**, 0.9997 for *Syrbatus superciliata* Asenjo & Valois **sp. nov.**, and 0.9930 for *Metopiellus crypticus*.

## DISCUSSION

The three new species described here of the Brazilian state of Minas Gerais state belong to *Syrbatus*, based on a sub-basal transverse sulcus on the base of pronotum (Figs. 2E, 3E and 4E), and lateral longitudinal sulcus on each side of the pronotum (Figs. 2E, 3E and 4E) (*Reitter, 1882*; *Park, 1942*). *Raffray (1904)* grouped the South American species of the genus in six species-groups based on morphological characters of head and antenna. Posteriorly, the concept groups were updated by *Park (1942)*, which were used here for the analysis and description of the three new Brazilian species of *Syrbatus* in the present work. In addition, to describe the new species with greater confidence, we found it necessary to study the other species of the species-group 2 (*Syrbatus brevispinus*, *Syrbatus bubalus*,
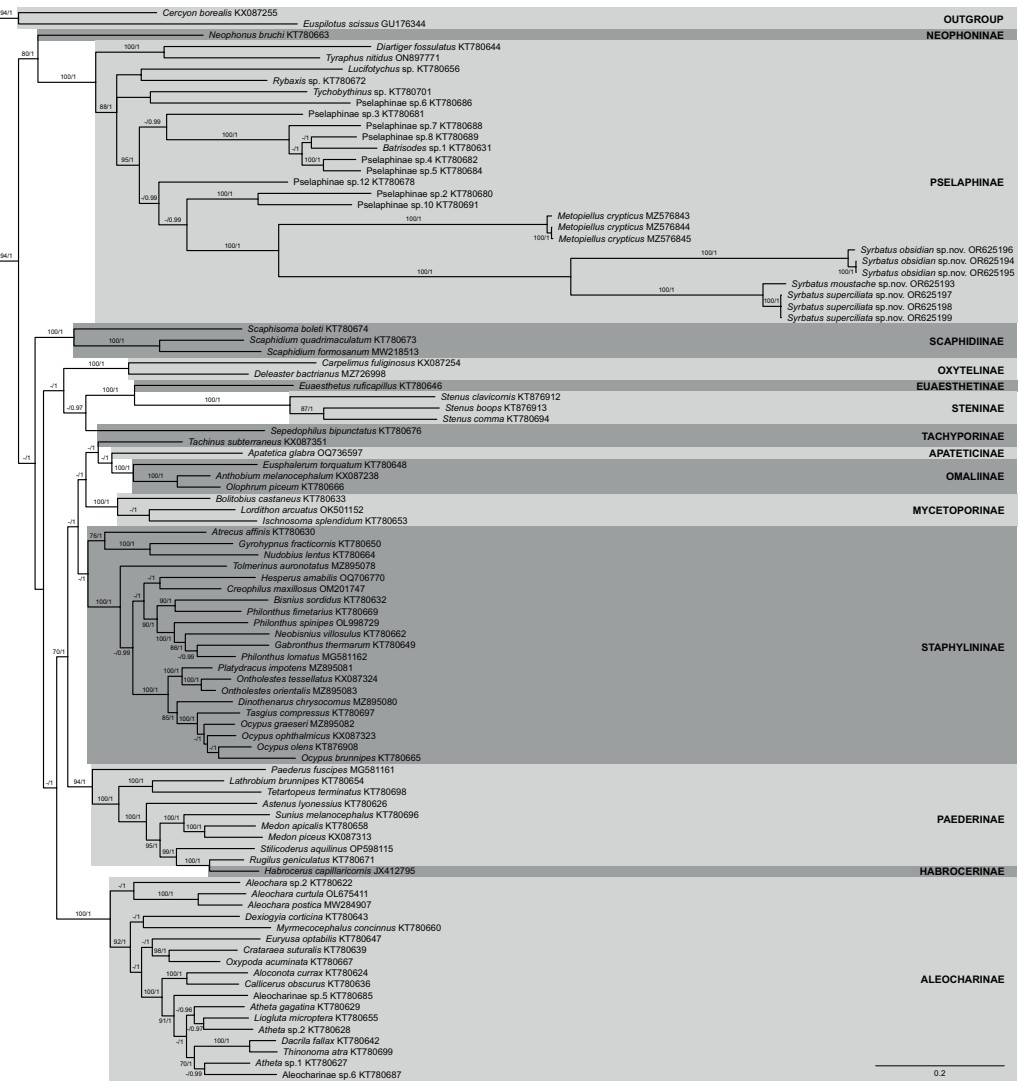

**Figure 7** **Phylogenetic placement of the three new species of *Syrbatus* within Staphylinidae, based on whole mitogenome data.** Majority-rule consensus phylogram of the Bayesian inference analysis evidencing the phylogenetic relationships among Staphylinidae species with available mitogenomes in the GenBank database and the three specimens of *Syrbatus moustache* Asenjo & Valois **sp. nov.**, *Syrbatus obsidian* Asenjo & Valois **sp. nov.** and *Syrbatus superciliata* Asenjo & Valois **sp. nov.**, indicating their respective GenBank accessions and subfamily affiliations. Statistical support values (BS ≥ 70 and PP ≥ 0.90) are indicated near the branches.

*Syrbatus centralis*, *Syrbatus grouvellei*, *Syrbatus hetschkoi*, *Syrbatus hiatusus* and *Syrbatus transversalis*), besides *Syrbatus trinodulus*, which is also known from Minas Gerais.

As mentioned above, *Syrbatus moustache* Asenjo & Valois **sp. nov.** and *Syrbatus superciliata* Asenjo & Valois **sp. nov.** were included in the species-group 2, characterized by the dorsally excavated head of the male (Figs. 2C and 4C), antennae simple (similar in both sexes; Figs. 2C and 4C), and genal region of the head not carinated (Figs. 2C and 4C). On the other hand, *Syrbatus obsidian* Asenjo & Valois **sp. nov.** was included in the

species-group 5, considering the genal area of the head longitudinally carinated (Fig. 3C), and antennae abnormal (Fig. 3B). Furthermore, the genital complex for female specimens of *Syrbatus* was studied, illustrated, and described for the first time (Fig. 5), showing its usefulness for the identification of the species with female sexual characters.

The three species described in the present study were found in caves inserted in siliciclastic rocks. These caves have between 10–500 m of horizontal projection, being inserted in all compartments of the landscape (low, medium, and high slopes) at altitudes that vary between 900–1100 m. In general, such caves are formed by talus deposits, with perennial underground drainage, and the presence of aphotic zones in the larger ones. However, the occurrence of *Syrbatus moustache* Asenjo & Valois **sp. nov.**, *Syrbatus obsidian* Asenjo & Valois **sp. nov.** and *Syrbatus superciliata* Asenjo & Valois **sp. nov.** in caves of small dimensions, and therefore without aphotic zones, besides the absence of evident troglomorphisms (*Christiansen, 2012*), suggests that the distribution of these new species may also be related to surface environments, characterizing the three of them as troglophiles.

The gene content and order tend to be highly conserved among the subdivisions of Insecta, with the structure of the ancestral insect mitogenomes being almost ubiquitous (*Cameron, 2014*). Nevertheless, some deviations from the standard configuration have been observed in the class, as in the case of some species of seed beetles (Chrysomelidae; Bruchinae), with long intergenic repeats and a minor reshuffling related to the gene trnQ (*Sayadi et al., 2017*). On the other hand, the three new species described here presented the mitochondrial genes in quite different positions, while all other currently known Staphylinidae mitogenomes share the standard insect gene order, as represented in the Fig. 6B. Additionally, a somewhat long intergenic region between trnS2 and ND1 was observed only for *Syrbatus obsidian* Asenjo & Valois **sp. nov.**, as another feature that differentiates this species from *Syrbatus moustache* Asenjo & Valois **sp. nov.** and *Syrbatus superciliata* Asenjo & Valois **sp. nov.**, besides the morphologic characters discussed above. Yet, the phylogenetic reconstruction with ML and BI, plus the high statistical support obtained in the ASAP and BPP analyses evidenced a robust delimitation among the three new species of *Syrbatus* described here.

Most of the sampled Staphylinidae subfamilies were recovered here as monophyletic and well supported in our phylogenetic reconstruction, at least in one of the two employed approaches, as, for instance, Staphylininae (BS = 68; PP = 1), which have been recovered only weakly supported in other analyses based on whole mitogenome data (*Song, Zhai & Zhang, 2021*; *Ji et al., 2023*). Considering the huge diversity of one of the currently known largest Metazoan families (*Grebennikov & Newton, 2009*; *Newton, 2022*), one should expect the retelling of phylogenetic history of Staphylinidae to be challenging. With its approximately 67,000 species (*Newton, 2022*), there are numerous difficulties in achieving relevant coverage and proportionality to reconstruct a robust molecular phylogeny of such an immense group. Depending on which taxa is included or left out, the resulting topology may vary considerably regarding recovered clades and support values, with genomic coverage and differences in evolutionary rates among used markers/genome portions also being an immense influence on the outcome(*e.g.*, *McKenna et al., 2015*;

*Kim et al., 2020*; *Motyka et al., 2021*; *Song, Zhai & Zhang, 2021*; *Ji et al., 2023*). Therefore, the divergences among topologies presented by different studies on the phylogenetic relationships within and among Staphylinidae subdivisions, even those employing the same set of markers/regions, such as whole mitogenomes, are not surprising. Yet, as the availability of genetic data for beetle taxa are continuously increasing, the causes of such conflicting phylogenetic signals may be clarified, allowing us to better address the many standing systematic issues of the infrafamilial relationships.

### Funding
This work was funded by Vale S.A. (Projeto Diversidade Biológica de Cavernas, R100603.CD.0X; Projeto Centro de Triagem de Invertebrados, R100603.CT.0X). Guilherme Oliveira is a CNPq (Conselho Nacional de Desenvolvimento Científico) fellow (307479/2016-1), also being funded by CNPq (444227/2018-0, 402756/2018-5, 307479/2016-1). The funders had no role in study design, data collection and analysis, decision to publish, or preparation of the manuscript.

### Grant Disclosures
The following grant information was disclosed by the authors:
Projeto Diversidade Biológica de Cavernas: R100603.CD.0X.
Projeto Centro de Triagem de Invertebrados: R100603.CT.0X.
CNPq (Conselho Nacional de Desenvolvimento Científico): 307479/2016-1.
CNPq: 444227/2018-0, 402756/2018-5, 307479/2016-1.

### Competing Interests
Guilherme Oliveira is an Academic Editor for PeerJ.

### Author Contributions
- Angélico Asenjo conceived and designed the experiments, analyzed the data, prepared figures and/or tables, authored or reviewed drafts of the article, and approved the final draft.
- Marcely Valois analyzed the data, prepared figures and/or tables, authored or reviewed drafts of the article, and approved the final draft.
- Robson de Almeida Zampaulo analyzed the data, prepared figures and/or tables, authored or reviewed drafts of the article, and approved the final draft.
- Michele Molina performed the experiments, analyzed the data, authored or reviewed drafts of the article, and approved the final draft.
- Renato R.M. Oliveira analyzed the data, authored or reviewed drafts of the article, and approved the final draft.
- Guilherme Oliveira conceived and designed the experiments, analyzed the data, authored or reviewed drafts of the article, and approved the final draft.

- Santelmo Vasconcelos conceived and designed the experiments, analyzed the data, prepared figures and/or tables, authored or reviewed drafts of the article, and approved the final draft.

## Field Study Permissions

The following information was supplied relating to field study approvals (*i.e.*, approving body and any reference numbers):

All studied specimens were collected in accordance with the sampling permits 065/2013 NUFAS/MG and 424.033/2018 and 49.994, granted by IBAMA/MMA and SEMAD/MG, respectively.

## DNA Deposition

The following information was supplied regarding the deposition of DNA sequences:

The assembled and annotated mitogenomes are available at GenBank: OR625193 to OR625199.

## Data Availability

The data is available at NCBI: PRJNA862473.

## New Species Registration

The following information was supplied regarding the registration of a newly described species:

Publication LSID: urn:lsid:zoobank.org:pub:2084B581-B904-486A-B237-0A9D9C839434.

*Syrbatus moustache* LSID: urn:lsid:zoobank.org:act:469E3B88-F2C7-4724-9F4C-7A51D1F5FABC;

*Syrbatus obsidian* LSID: urn:lsid:zoobank.org:act:C42D432B-4893-4324-A859-1BD40E47E6D7;

*Syrbatus superciliata* LSID: urn:lsid:zoobank.org:act:489AB8BC-FF86-4214-AC28-850C99EA37B1.

## Supplemental Information

Supplemental information for this article can be found online at http://dx.doi.org/10.7717/peerj.17783#supplemental-information.

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
