# Peer review of "New taxonomic insights for Brazilian Syrbatus Reitter (Coleoptera: Staphylinidae: Pselaphinae), including three new species and their mitochondrial genomes"

_PeerJ, doi:10.7717/peerj.17783_

## Round 0.1 · original submission · Minor Revisions

Please, take all the reviewers' comments and suggestions into account, including those that are provided directly in the text.

Reviewer 1 ·

Basic reporting

See general comment.

Experimental design

See general comment.

Validity of the findings

See general comment.

Additional comments

All my comments are attached to the PDF. There are several major points that need to be stated:

1) The gender of Syrbatus is masculine, the new species names are either not Latinized or feminine.
2) The diagnosis section should include a list of diagnostic characters of the new taxon, not a comparison. The content now included in the Diagnosis should be moved to a new section, e.g., 'Comparative notes'.
3) The 'Mitogenomes and phylogenetic placement' contributes little to the phylogeny of Syrbatus nor the taxonomic core of the paper, and therefore should be deleted or removed to the supplement file as a whole.

Annotated reviews are not available for download in order to protect the identity of reviewers who chose to remain anonymous.

·

Basic reporting

Overall, it is a well written paper. While there are a few minor errors and areas that could be rephrased for enhanced clarity, these have been highlighted in the attached PDF along with suggested revisions. The paper boasts a well-organized structure, incorporates pertinent figures of high quality, and provides ample raw data in the supplementary material. Consequently, it is almost issue free in this regard.

Experimental design

The methods employed throughout all sections, encompassing both morphology and molecular data processing, are appropriate. However, I would like to address a concern pertaining to the approach taken in species description. It appears that the authors focus on describing the holotype rather than providing a comprehensive depiction of the entire species. While this is not inherently incorrect, considering the flexibility granted by the code, I respect the authors' decision, albeit finding it somewhat philosophically flawed in terms of understanding the ontological constitution of a holotype.

To enhance the biological information conveyed in the paper, I recommend the following:

1. Although there is a table of measurements provided in the supplementary material, the text exclusively features individual measurements for the holotypes alone. Emphasizing the range in the main text is crucial from a biological perspective, surpassing the significance of measurements from a single specimen, regardless of its type status;

2. The authors have included a section discussing differences between the holotype (male) and females, which makes the description of Syrbatus moustache and Syrbatus superciliata, each having just a male (holotype) and two females (paratypes), less of a concern. However, for Syrbatus obsidian, featuring seven males (including holotype) and four females (paratypes), providing information on just one male specimen (holotype) and comparing it to females, in my view, is insufficient. I suggest the authors include a section on the variation across all specimens, encompassing not only the range of measurements (as suggested previously, and somewhat already included indirectly in the supplementary material) but also variations in the morphological characters provided in the holotype's description.


Again, it is not mandatory, but I urge the authors to consider this suggestion to improve the clarity of the biological information conveyed in the paper.

Validity of the findings

The authors have described three new pselaphine species within the genus Syrbatus, discovered in caves in Brazil. Additionally, they have assigned lectotypes for certain other Brazilian species and identified holotypes for other three. In conjunction with the morphological descriptions, the authors have provided the mitochondrial genomes for all three newly described species. Their thorough investigation includes a phylogenetic analysis aimed at understanding the placement of these species within Staphylinidae; which also reveals a robust molecular delimitation among the three species in study.
The results are commendably presented, and data-rich. I extend my congratulations to the authors for their dedicated work and the evident effort invested in this research.

Reviewer 3 ·

Basic reporting

While the overall English writing is well-executed, there are some issues with the use of morphological terms and explanations in the manuscript. I have provided comments on areas where these issues were identified. Concerning the figures, the provided figures maintain the clarity required for understanding traits in a taxonomic paper. However, some figures, for example head, are not helpful for understanding the characteristics such as the character of the oval transversal area in the head. The adjustments in contrast and brightness, or the inclusion of sketches, enhance the reader of this paper to understand the morphological character. Moreover, the absence of orientation explanations for the photos of the Aedeagus in each figure is unkindness. To understand the highly complex Aedeagal morphology of the supertribe Batrisitae, providing clear illustrations is crucial for future taxonomic considerations.

Experimental design

The experimental methods are clearly and meticulously described. This study, which analyzed the mitochondrial genome and identified significant variations in the composition of mitochondrial genome regions for the newly described species, presents valuable results. However, regrettably, no other novelty beyond the mitochondrial genome analysis is apparent. Additionally, the detailed reasons for elucidating the systematic positions of the three newly described species are not provided. it would be advisable to provide a detailed explanation of why phylogenetic analysis was made for these three new species. Personally, I believe it would have been valuable to verify the species-group classification using genetic information.

Validity of the findings

The species diversity of the subfamily Pselaphinae in tropical regions, particularly in the Neotropical region, is still poorly studied. Therefore, the addition of new species with detailed descriptions is valuable. Additionally, the inclusion of clear photographs of holotype specimens of known species enhances the usefulness of the paper. In describing new species, the comparison with known species is crucial, and this paper will be beneficial for future taxonomic studies. The designation of holotype and lectotype is also greatly appreciated. However, the absence of the Keys for species-group and/or species of the genus Syrbatus from Brazil was regrettable.
While the paper is beneficial in promoting mitochondrial genome analysis in the subfamily Pselaphinae, the novelty of the results is a little evident. Additionally, 1/3 part of the discussion of phylogenetic analysis focused on the overall phylogenetic relationships within the family Staphylinidae (which is useful in itself), but it seems somewhat divergent from the aim of this paper.

Additional comments

・Please use an en-dash to indicate the range.
・The statistical support values presented in the phylogenetic tree differ from those explained in the results. Please accurately specify the Statistical support values in the figure.
・The diagnosis of the genus Syrbatus and similar genera is unclear. Please provide a detailed diagnosis in the introduction.

Annotated reviews are not available for download in order to protect the identity of reviewers who chose to remain anonymous.

---

## Round 0.2 · Minor Revisions

Please, address minor suggestions by Reviewer 3 regarding the information on the phylogenetic analysis and morphological characters. After that, I would be happy to accept your manuscript for publication.

Reviewer 1 ·

Basic reporting

no comment

Experimental design

no comment

Validity of the findings

no comment

Additional comments

After an extensive revision, the paper can be accepted in its current form.

·

Basic reporting

I maintain my initial review that the paper is well-written, with a well-organized structure, relevant figures, and essential raw data provided in the supplementary material. The authors have addressed the few minor issues in this new version, and all suggestions were thoughtfully considered, with the text modified accordingly.

Experimental design

The contradictory points I highlighted in the previous version regarding the examined material have been addressed in this new version. As I pointed out in the last review, the methodology for both the morphology and molecular aspects of the study remains appropriate.

Validity of the findings

Everything I previously mentioned regarding the significance of this work still stands, and I have nothing further to add in that regard. Once again, I extend my congratulations to the authors for their contribution and hard work on this paper.

Reviewer 3 ·

Basic reporting

I think the content has been improved. A few minor points were noted for your reference. One point is that the introduction does not indicate why phylogenetic analysis is being done and why mitogenomes are being used. I think the phylogenetic issue should be clearly stated in the phylogenetic paper.

Experimental design

The experimental methods are clearly and meticulously described.

Validity of the findings

It is the same as the first peer review. It is useful as a taxonomic paper. Can you add the Diagnosis of each species-group to the end of the taxonomy section?
For example, following style…

Species-group 1
S. xxxx sp 1
S. xxxx sp 2

Diagnosis. This species group can be distinguished by the following characteristics: head ...

Additional comments

If female specimens are included in the molecular analysis, the sexes can be identified based on the gene information. You may add the results into the description of females. However, if females are not included in your analysis, you can disregard this comment.

Annotated reviews are not available for download in order to protect the identity of reviewers who chose to remain anonymous.

---

## Round 0.3 · accepted · Accept

Thank you for addressing all reviewers' concerns. I think the current form of the manuscript is ready for publication.